# Improving early prediction of crop yield in Spanish olive groves using satellite imagery and machine learning

**M. Isabel Ramos**[1]*, **Juan J. Cubillas**[2], **Ruth M. Córdoba**[3], **Lidia M. Ortega**[4]

**1** Department of Cartographic, Geodesic and Photogrammetry Engineering, University of Jaén, Jaén, Spain, **2** Department of Information and Communication Technologies Applied to Education, International University of La Rioja, Logroño, Spain, **3** Master's Degree student in Geomatic Engineering and Geoinformation, University of Jaén, Jaén, Spain, **4** Department of Computer Science, University of Jaén, Jaén, Spain

* miramos@ujaen.es

**Data Availability Statement:** All Supporting Information files files are available from the ZENODO database (accession number(s) 10.5281/zenodo.11102805).

## Abstract

In the production sector, the usefulness of predictive systems as a tool for management and decision-making is well known. In the agricultural sector, a correct economic balance of the farm depends on making the right decisions. For this purpose, having information in advance on crop yields is an extraordinary help. Numerous predictive models infer accurate crop yield data from aerobiological variables and pollen analysis; this is around spring, in the middle stage of the crop year, when planning and investments are already done. The aim of this study is to anticipate accurate crop yield data at an early stage of the cropping season. In the case of olive groves in Spain, this period is in February. This work is developed for an entire province, Jaen, belonging to the region of Andalusia, in Southern Spain. The methodology uses Machine Learning algorithms together with an exhaustive analysis of predictor variables. Temporal data come from public web services, such as spatial data infrastructures of some state agencies. The processing of the satellite imagery is carried out by the geospatial processing service Google Earth Engine. The result is the early prediction of kilograms of both olive crop and olive oil, eight months prior to the beginning of the first harvesting campaigns of the year, with an average absolute error of prediction better than 26%. The relevance of this work is the early availability of predicted crop yield data together with the multi-scale applicability of the predictive models. This makes this model a useful tool for all the agents involved in olive grove management. From farmers, to agricultural technicians, researchers and scientists dedicated to the study of the olive tree, to governmental institutions and agricultural associations that provide technical support, advice and regulation to ensure responsible practices and the long-term viability of the olive industry.

## Introduction

Agriculture continues to be one of the occupations that contribute most to the economy of many countries in the world. Even though worldwide agriculture impacts in the Gross

**Funding:** This research has been partially funded through the research projects HORIZON-MISS-2021-SOIL-02-03 and HORIZON-MISS-2023-SOIL-01, which are financed with the European Union's Horizon Europe research and innovation programme awarded to LMO, RMC and MIR. We are also grateful for the support provided by the Ministry of Innovation and Science of the Government of Spain through the research projects TED2021-132120B-I00, PID2021-126339OB-I00, PID2022-137938OA-I00 and GOPO-JA-23-0008 awarded to MIR, LMO and JJC. Finally, the research project GOPO-JA-23-0008 which is financed by the European Union FEDER funds has also contributed to the financing of this work awarded to MIR and LMO. The funders played a part in the decision to publish.

**Competing interests:** The authors have declared that no competing interests exist.

Domestic Product (GDP) about 6%, crop productivity plays an important role in specific areas, such as the Mediterranean basin. The climate of the countries of the Mediterranean area has conditioned the type of agriculture, focusing mainly on vines, cereals and olives.

Spain is the largest producer of olive oil in the world, and within this country, the province of Jaen (in the south of the country) is the most productive. This is the main economic activity in the area. According to data published by the Spanish Ministry of Agriculture, Fisheries and Food (MAPA) in 2020, the province of Jaen (2,67% of the national territory) represents almost 24% of the olive grove area in Spain [1].

The productivity of crops, including the olive groves (Olea Europaea L.) depends on various factors such as soil fertility, weather condition, the type of tillage performed and also the amount of harvest of the last year. Some of these factors are controllable by the farmer, but others cannot be acted upon directly. This is the case of the climatic conditions of recent years, which are considered to be the main determining factors for the drop of more than half in production in the last two seasons. This imbalance has not occurred to the same extent in other countries such as Greece. However, Spain by far is the largest EU exporter of olive oil to the rest of the world, and this has led to tension in the international olive oil market, pushing up prices. This has increased olive oil prices up to three times in just four years in the country.

Good crops are a fundamental objective for farmers, but also for oil mills, distribution and insurance companies, as well as the agri-food industry. Early crop forecasting is very important to know the future behavior of the markets and it can be considered a hot topic in the scientific literature. In olive growing, *early prediction* is considered to be that which precedes pollination, that is, in the months of February and March (in the northern hemisphere). In the particular case of the olive grove, most of the work in the literature focuses on the study of physical measurable variables such as the amount of pollen in the air, vegetative indices that indicate plant health, or rainfall maps. However, this prediction is considered *late prediction*, since these variables are captured in springtime between April and May, when there are already other visible features that allow agricultural technicians to predict how abundant the next harvest will be. On the contrary, an early prediction cannot make use of these directly observable variables, but uses the historical behavior of the place (at the municipality level) and the environmental and climatic variables of the immediately preceding months. This leads us to predict the behavior of the next harvest approximately because not all the factors that will affect are known at this stage. However, it is possible to advance what the general trend will be. In fact, some other factors are decisive during the following months: the processes of flower pollination, the fruit setting and its subsequent ripening. However, the behavioral trend of the next crop can be obtained with controlled error with the regression techniques of our model. In any case, this predictive model will be fed as the seasons progress to better adjust to the time of harvest.

Summarizing, this work focus on the early crop yield prediction based on several representative variables such as: (1) the previous year's production, (2) rainfall at specific times, (3) other meteorological factors in the previous autumn and winter, (4) as well as the most relevant vegetation indices. The study is made at municipality level, scale in which it is possible to use satellite information on the variables mentioned above. But we only perform calculations in the polygons of plots where an olive plantation is known to exist. This early forecasting process has been carried out throughout all the municipalities of the province of Jaen, the area with the highest concentration of olive groves on the planet. Note that each municipality is made up of several olive groves.

The process has been carried out using Machine Learning (ML) techniques, a branch of the Artificial Intelligence (AI) capable of making an early prediction in many research fields. By using Machine Learning techniques, it is possible to determine patterns and correlations from

different types of variables and discover knowledge from these datasets. Our case belongs to the category of supervised learning problems. This means that we have the values of the input predictors for each output value. In supervised machine learning, training algorithms fall into two main categories: Classification and Regression. Classification deals with categorical data and Regression deals with purely numerical data. This separation of nature in the data can refer to both input and output variables. In our case we have focused on the usefulness of the predictive measure, olive crop yield, which must be a numerical value. For this reason, regression algorithms have been applied in this work. The system has been trained by collecting the values of all input and output variables over the course of a decade. The variables to be predicted are both the quantity of kilograms of olives and the quantity of oil obtained. The difference between these two values is known as the crop yield, and can vary from season to season.

The methodology proposed works in two phases. In a fist phase, the model is trained with historical data, allowing to properly determine this model, since the results are obtained from past experience. In the testing phase, part of the historical dataset is used for training, and the rest for evaluating performance. The olive crop yield is understood as the two variables described above, the number of kilograms of olives to be harvested and the amount of olive oil obtained. It is necessary to understand that both variables are related but not in direct proportionality.

The paper is structured as follows. The following section provides an overview of the state of the art regarding the techniques and input data used in this early prediction study. The materials and methods section describes the area covered by the study and the characteristics of its olive plantations, as well as the spectral information and vegetation indices. The workflow of the developed methodology is also provided. This is followed by an analysis of the results of the different tests performed, justifying the reasons for each of them and deepening the knowledge offered by the resulting data. Finally, the conclusions section gathers the main findings of this work and proposes future applications.

## State of the art

The significant drop in production in the last two seasons compared to the previous ones is the main reason for the increase in olive oil prices. From the 1500 thousand of tons in 2021/22 to 666 in 22/23 and finally 853 thousands in the last crop season (23/24) [2]. Therefore, any yield prediction relating to subsequent harvests are very important for the companies in the sector and consumers. Most crop forecasting systems in olive production are done in the spring. However, advance this prediction to the beginning of the year has numerous advantages in the economy:

- **Farmers:** Many families in Jaen and the rest of the Mediterranean basin derive their income from agriculture. An early forecast allows them to adjust their household finances and anticipate the hiring of new personnel.

- **Agricultural cooperatives and oil mills:** In the area under study, it is common for farmers to form cooperatives. They guarantee the milling process, sale of the product and all kinds of consulting services. For all of them, an early forecasting allows to better adjust the prices at which they market the product. It can help to know in advance the personnel to be hired, or to improve facilities to face a successful campaign.

- **Insurance companies:** These companies move a lot of money and are particularly interested in knowing the risks and adjust premium prices or indemnities in order to establish their rates.

- **The food and distribution industry:** Olive oil is one of the most appreciated oils because of its culinary and healthy properties. These companies have to make decisions about the most advantageous moment to market the oil.

- **The international market:** Spain is the world's largest exporter of olive oil, with 65% of its total sales. Knowing its production in relation to other producing countries allows to establish better market strategies.

- **The agricultural machinery industry sector:** Over the past decade, the olive industry has experienced a significant surge in mechanization. This mechanization primarily targets the reduction of harvesting expenses, which represent the highest cost within olive farming. Early forecasting makes it possible to better adjust machine production to actual demand in advance.

In view of the above, the advance in the prediction process is challenging from an economic as well as a scientific point of view. From an environmental point of view, the implications are also relevant. This study bases its prediction on the study of olive grove health through vegetation indices. This knowledge makes it possible to monitor its condition and adjust specific actions based on improving the initial prediction values, avoiding the indiscriminate use of fertilizers or pesticides.

Crop prediction is key for many growing types. Although, some of them such as olive groves, have a long vegetative cycle with numerous factors affecting pollination and fruit set. Scientists and farmers strive to identify these and numerous other variables to better understand their impact on crop production.

In the past, crop predictions heavily relied on farmers' expertise, but today, much of agricultural research centers on improving crop forecasting. This shift has been complemented by other economic sectors using the predictive potential of Artificial Intelligence (AI) techniques, specifically Machine Learning (ML) and Deep Learning (DL). These techniques are now extensively employed for crop yield prediction [3–6]. This helps the olive sector to make decisions for the best management strategies.

DL or ML regression techniques are mostly used for determining the degree to which independent variables influence dependent variables. In the particular case of the olive grove, it is possible to establish a relationship between *crop production* (dependent variable) and other previously referred variables, such as vegetation indices, temperatures or irrigation [7, 8], plant health [9], the previous harvest, even the terrain slope or the age of the olive trees [10] (the independent variables). There is undoubtedly a correlation between these factors. Years of abundant crops are typically marked by well-distributed rainfall throughout the fruit development stages, adequate temperatures and a moderate yield form the previous year. Production is obviously also related to the agricultural practices employed by farmers, encompassing activities such as tree pruning, the application of fertilizers, and the use of various phytosanitary products aimed at managing pests and weeds.

In this sense, most works related with crop prediction applies regression techniques to some of these factors independently or through a combination of them. One of the most studied variables is the amount of pollen in the air [11–13]. Although there is a direct relationship between pollen quantity and crop yield, this factor alone is not decisive, as it also affects other factors such as water deficit, extreme temperatures or phytopathological problems between flowering and harvest [14]. The most reliable prediction occurs once fruit set has begun, what happens after pollination is complete. However, this delays the prediction process too much. In fact, agricultural technicians are able to sample branches and count fruit by hand. Extrapolating these values, it is feasible for them to make a forecast of the next crop in the farm.

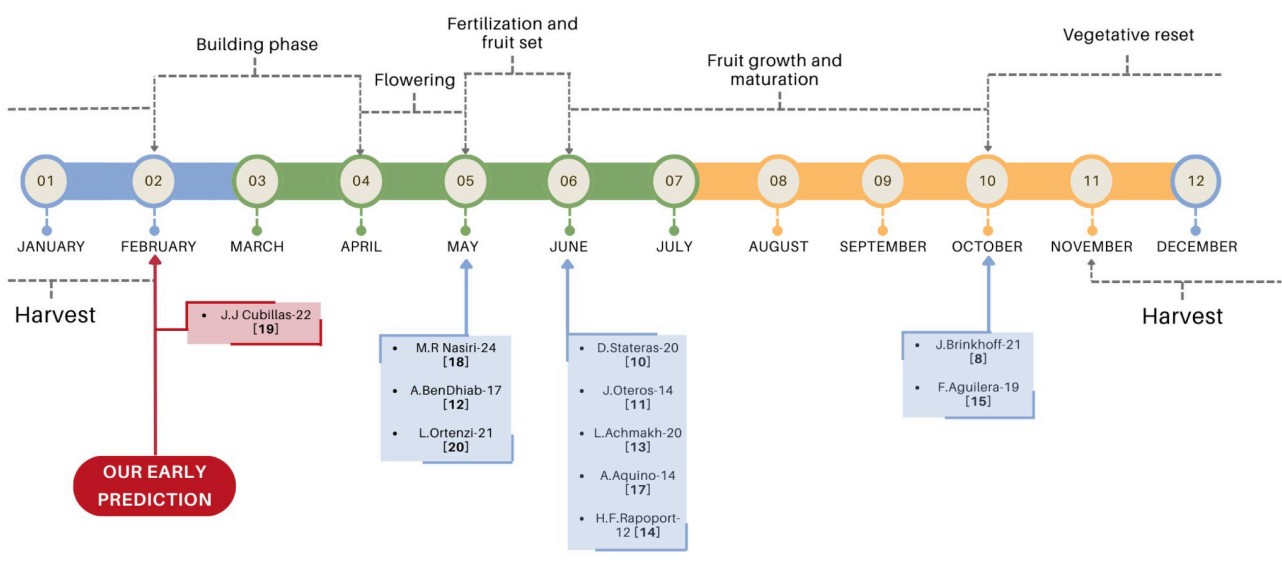

**Fig 1.**

Therefore, a late prediction may be unnecessary as it does not provide much information on a small scale.

Most of the literature consulted is *late prediction*, performed only five months before the next crop [12, 15]. In [16] it is introduced a conceptual model for predicting fruit oil content in order to establish the optimal harvesting period, but it takes place even later. Moreover, recent works, such as [17], predict olive fruit with a low predictive error, 2.64%, but this data is provided from computer analysis of data visible in the field, i.e. two months prior to fruit harvesting. Others as [18] perform fruit prediction from tree root analysis. In this case, it is necessary the installation of sensors in the farm and take several samples throughout the year. However, our work makes the prediction eight months, for the early harvest, from the beginning of the first campaigns of the year, with remote and non-invasive techniques.

In fact, only a few papers are really focused on early estimation in the literature, but some of them actually considering the estimate in spring months instead of the winter prior to harvesting [19]. Fig 1 reflects the phenology of the olive grove together with the annual chronology in which the bibliography related to olive crop yield prediction performs its predictions calculations. Except in the work Cubillas et al [20] whose prediction is before flowering, the rest of studies can be considered late prediction. Although this referred paper only works at the farm level and without using spectral information. Therefore, this paper is the first study on *early crop forecast* considering the most representative variables according to the related bibliography and the agricultural technicians consulted: vegetation indices, climatic conditions and the previous harvest information. We now analyze the importance of these variables in our study and in the literature.

## Spectral information and vegetation indices

Remote sensing applications provide a variety of information on all types of crops. In the last years, an increasing source of information is provided by unmanned aerial vehicle (UAVs) platforms, mainly equipped with optical multispectral cameras, to map, monitor, and analyze, temporal and spatial variations of vegetation using spectral vegetation indices (VIs). However, for this study, historical data series were not available from this source.

Satellite imagery is also widely used despite spatial resolution is not adequate to capture the surface variability required in many precision agricultural systems. However, they are increasingly improving temporal and spatial resolution, being sufficiently precise to carry out studies at municipality level, as is the case in this work. The advantages are accentuated by their accessibility, larger area coverage, less processing and operator interaction coupled with more spectral information. In any case, satellites and airborne multispectral cameras are not opposing technologies, they can be used together as complementary data.

Multispectral and hyperspectral sensors generate images consisting of multiple band wavelengths. Linear combinations of them, the so-called vegetation indices, are aimed at highlighting relevant features about plants and soil. In particular, a total of 82 spectral vegetation indices were collected from the literature by Pu, Ruiliang [21]. Between this index set, the most widely used is the Normalized Difference vegetation Index (NDVI), whose relationship with olive grove production has already been reported [22]. Literature has studied the relationship of this index and the use of fertilizer in olive groves [23, 24], water deficit and antioxidative enzyme activity [25]. The fact that the index behaves differently depending on the type of vegetation, has served to differentiate each of the studied crops [26].

In addition to the NDVI index, some studies work with a combination of some others to determine which of them perform better for different crops [27, 28]. Table 1 provides information about a selection of these and other indices in the scope of olive grove studies. It has been reviewed which vegetation indices are related to vegetation vigor in [9, 29, 30], including indices such as NDRE or SAVI between others. Water stress has been related to GRVI, MSI and NDMI indices [8, 31–33], fertilization with the GNDVI [24] or vegetation density [34] using GCI. On the contrary, some well known moisture-related indices such as the Normalized Difference Moisture Index (NDMI), have not been related so far to crop prediction in olive groves, but it will be considered in our study because of the importance of moisture in crop yield.

As the first step in our methodology, these indices included in the bibliography have been checked in order to determine their relationship with crop prediction. Some of them have been discarded, and on the contrary, some others have been added to determine which of them most contribute for forecasting in the particular olive grove case.

## Meteorological variables

Crop yields depend to a great extent in climatic conditions. As crop conditions are being affected by climate change, some studies are being conducted to study the viability of long-term crops in relation with the geographic location [35].

**Table 1. List of vegetation indices included in the literature related to olive cultivation.**

| Index | Index name | Formula | Bibliography |
|---|---|---|---|
| NDVI | Normalized difference vegetation index | $\frac{NIR-Red}{NIR+Red}$ | Used in most papers |
| GNDVI | Green normalized difference veg. ind. | $\frac{NIR-Green}{NIR+Green}$ | [9, 24, 28, 29,34] |
| MSI | Moisture Stress Index | $\frac{NIR}{SWIR}$ | [31–33] |
| NDMI | Normalized Difference Moisture Index | $\frac{NIR-SWIR}{NIR+SWIR}$ | - |
| NDRE | Normalized difference vegetation index | $\frac{NIR-RedEdge}{NIR+RedEdge}$ | [9, 29, 30, 34] |
| GRVI | Green Red Vegetation Index | $\frac{Green-Red}{Green+Red}$ | [8] |
| GCI | Green Chlorophyll Index | $\frac{NIR}{Green} - 1$ | [28] |
| SAVI | Soil Adjusted Vegetation Index | $\frac{NIR-Red}{NIR+Red+L}*(1+L)$ $0 <= L <= 1$ | [9, 29, 30, 34] |

The olive grove is perfectly adapted to the Mediterranean climate and the plant usually survives adverse meteorological situations. However, in adverse situations, the plant ceases to produce. Production is definitely determined by rainfall (or irrigation) and temperatures [36–38]. In fact, low rainfall conditions and high temperatures are considered to be the most important factors influencing the production decline for the 2022/23 and 2023/24 seasons in Spain.

ML and DL regression algorithms are traditionally used to correlate meteorological data with the harvests obtained [39]. The capture of this information can be performed both using meteorological stations or satellite data. In [40], the use of satellite data from Copernicus' ERA5 for temperature measurements is advocated for modeling phenological phases of olive orchards, providing better predictions in comparison with weather stations.

Meteorological variables are widely used in regression algorithms for the olive grove sector [12, 41]. Differentiating between climate variations by seasons is significant since it affects the different vegetative stages of the plant and the fruit [15]. In [42] this influence is related to the presence of the olive fruit fly, which is one of the pests that most affects this crop. In any case, rainfall is also a relevant factor in areas with shortages, such as the Andalusia region in southern Spain. In fact, drought effects on vegetative health and its effect is detected by analyzing vegetation indices, such as NDVI, GRVI [8], NDWI (Normalized Difference Water Index), NDMI (Normalized Difference Moisture Index), MSI or SAVI [4, 31].

Drought problems occurs even in irrigated crops. Irrigation makes it possible to mitigate water shortages at critical moments, such as flowering, fruit set and stone hardening (the first stage in the biological formation of the oil in olive fruits). In these cases, a water deficit would cause the fruit to fall. Water shortage phases also affect the reservoirs, and that leads to sparse irrigation aimed at saving the plant, rather than the crop. Olive groves need at least 0.4 $m^3$ per square meter per year, and correctly spread over time, otherwise the plant may survive but reduce its production.

Water stress can be detected by specific vegetation indices [8, 27, 32]. This analysis is usually focused on in the spectral response of the soil with indices such as NDMI (Normalized Difference Moisture Index) or SAVI (Soil Adjusted Vegetation Index), this last used to detect the influence of soil brightness in areas where vegetative cover is low.

Hyperspectral imaging surpasses multispectral cameras in resolution by obtaining a much larger number of spectral bands. This technology can anticipate drought stress by analyzing enzyme activity, photosynthetic rates [25] or the leaf water potentials by analyzing specific vegetation indices [31]. In any case, climate is not the only variable to be considered, as other favorable conditions for a good crop have to be considered and correlated, as it is the plant's health [43].

Despite all the research work, only a few studies really focus on determining which set of combined factors are relevant for this predictive study. In [44, 45] climatic variables along time and VIs are jointly used for crop forecasting. Some recent works obtain the information about meteorological variables, such as soil moisture, by analyzing specific vegetation indices [32]. It is common to face an inherent problem of merging data from different sources, considering variable accuracy and scale levels between them [28].

## Material and methods

### Study area

The study area comprises all the municipalities of the province of Jaen, extending from 38°N 3°W (ETRS89) in Andalusia, southern Spain. The province of Jaen spans a total area of 13496 $km^2$, encompassing diverse topographies ranging from mountainous regions to extensive valleys, with a total of 97 municipalities. The climate is Mediterranean, and the average annual

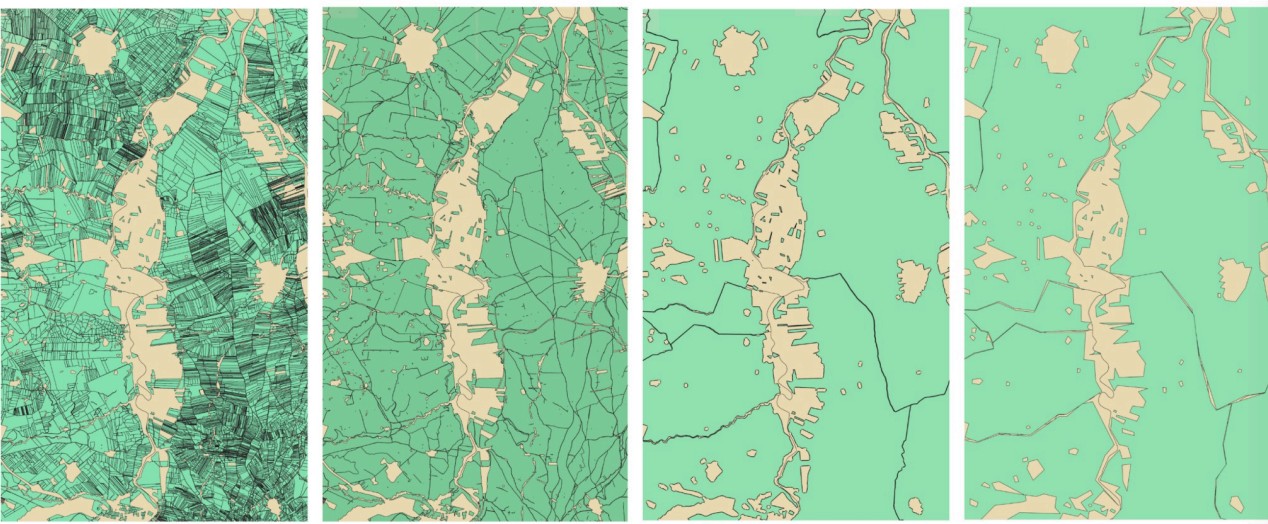

**Fig 2.**

temperature is 15-17˚C with a wide temperature range between day and night. Rainfall is around 500 mm and accumulates in the coldest period, although this factor varies according to the orography of the area.

Knowing these characteristics, the data obtained including rainfall, temperature and vegetation index values are associated with each of the municipalities. This information is aggregated by seasons and municipalities to obtain averages from each one.

Although a significant portion of the province of Jaen is dedicated to olive cultivation, we distinguish among the different land uses in order to identify only those devoted to olive crop. Then, all calculations are made using data of these selected areas, avoiding the rest of areas such as population centers, reservoirs, ponds, rivers, and road networks mottling the province. This greatly reduces noise in the input data.

We use only public and free accessed information about olive tree plots from the SIGPAC [46], a Geographical Information System for the Identification of Agricultural Parcels, which is maintained by the competent agricultural Delegation of the Andalusian Government. SIG-PAC contains updated information until 2023.

In the province of Jaen is noticeable the amount of land occupied by this type of cultivation, as well as the administrative division of municipalities. It is worth noting the great difference between olive groves located in the mountains, which occupy marginal soils, and modern high-intensity plantations located in areas with more favorable agronomic characteristics. Between the two, average yields can vary by up to 30 times the total volume (500 kg/ha/year versus 15000 kg/ha/year) [47]. The selection of the different land uses has been performed with QGIS [48] software, in which only those plots of olive trees have been considered. Later, this vector information is uploaded to Google Earth Engine to process the spectral images, and extract only the information related to these areas. In Fig 2, it is possible to observe the simplification process from the first data obtained. This involves generalizing the polygons that define the outline of each municipality to simplify their geometry as much as possible in order to lighten the volume of the shapefile.

This tool allows to perform massive calculations in an agile way, and as the study is done at the scale of a municipality, small details are not compromised.

## Dataset

The fundamental basis of any predictive study is the availability of reliable data to ensure a quality prediction. In this case study, the dataset consists of information from official sources and agencies, guaranteeing data reliability and free access. This accuracy in the results sometimes involves the need for some pre-processing such as, outlier removal, data cleaning, classification, data normalization, etc.

The olive tree has a vegetative cycle of two growth stages in which the care and feeding of the plants is essential to guarantee a good yield after harvesting. The olive trees go through a phase of budding (February-April), flowering (April-May), fertilization and fruit set (May-June), fruit growth (June-September), fruit ripening (October-December) and vegetative rest (November-February). In this work, these stages are grouped into three, namely seasons, which coincide with the times of the year when the olive tree responds uniformly to the external factors acting on it. Table 2 shows the months that include each season considered.

In general terms, the yield of the olive grove depends on many variables. However, this study is conducted at municipal aggregation level. Therefore, we consider only those factors that are not inherently embedded in the harvest data or the municipality's geographical context. These are the quality of the soil, the presence of pests, tillage, olive tree varieties and planting density. On the contrary, environmental and meteorological variables changes behavior from one year to the next, as well as those variables that indicated the health of the crop during the crucial seasons. These are certain vegetation indices that are calculated from satellite images. In fact, the influence of data differs depending on the two objectives of the study, the kilograms of olive crops or kilograms of olive oil. Furthermore, the impact of the same data on the predicted target may vary based on their values at a particular date.

**Olive crop yield data.** The Regional Government of Andalusia publishes annual olive crop yield data at different levels of aggregation. We obtain this information grouped at each of these municipalities, encompassing a dataset spanning fifteen years, from 2009 to 2022. Crop yield is understood as a double variable, on the one hand as the quantity of kilograms of olives harvested annually, and on the other hand, as the kilograms of oil extracted. The quantity of oil obtained varies depending on the quality of the olives harvested. Not all olive crops produce the same amount of olive oil. Even though the two targets of this research (Kg of harvested olives and Kg of oil) have a certain relationship of dependence, the dataset influences each one in a different way. For this reason, a corresponding predictive model is generated for each one.

**Vegetation indices.** Vegetation indices are calculated from specific spectral ranges from multispectral imagery. Even though Sentinel-2 provides better resolution, we finally have combined Landsat 7 and 8, as the former has been available since June 2015 and our study attempts to study the behavior of the crops further back in time from 2009. With the use of Landsat 7 and 8, all the vegetation data possible are obtained, counting with the limitations of said satellites, such as their spatial and temporal resolution. Both satellites has a resolution of 30m and their frequency is of 16 days, but Landsat-7 working from 1999 and Landsat-8 from 2013.

**Table 2. Season division.**

| Year | Dates |
|:---:|:---:|
| Winter | 01-December to 29-February |
| Spring | 01-March to 30—June |
| Summer—Autumn | 01-July to 30-November |

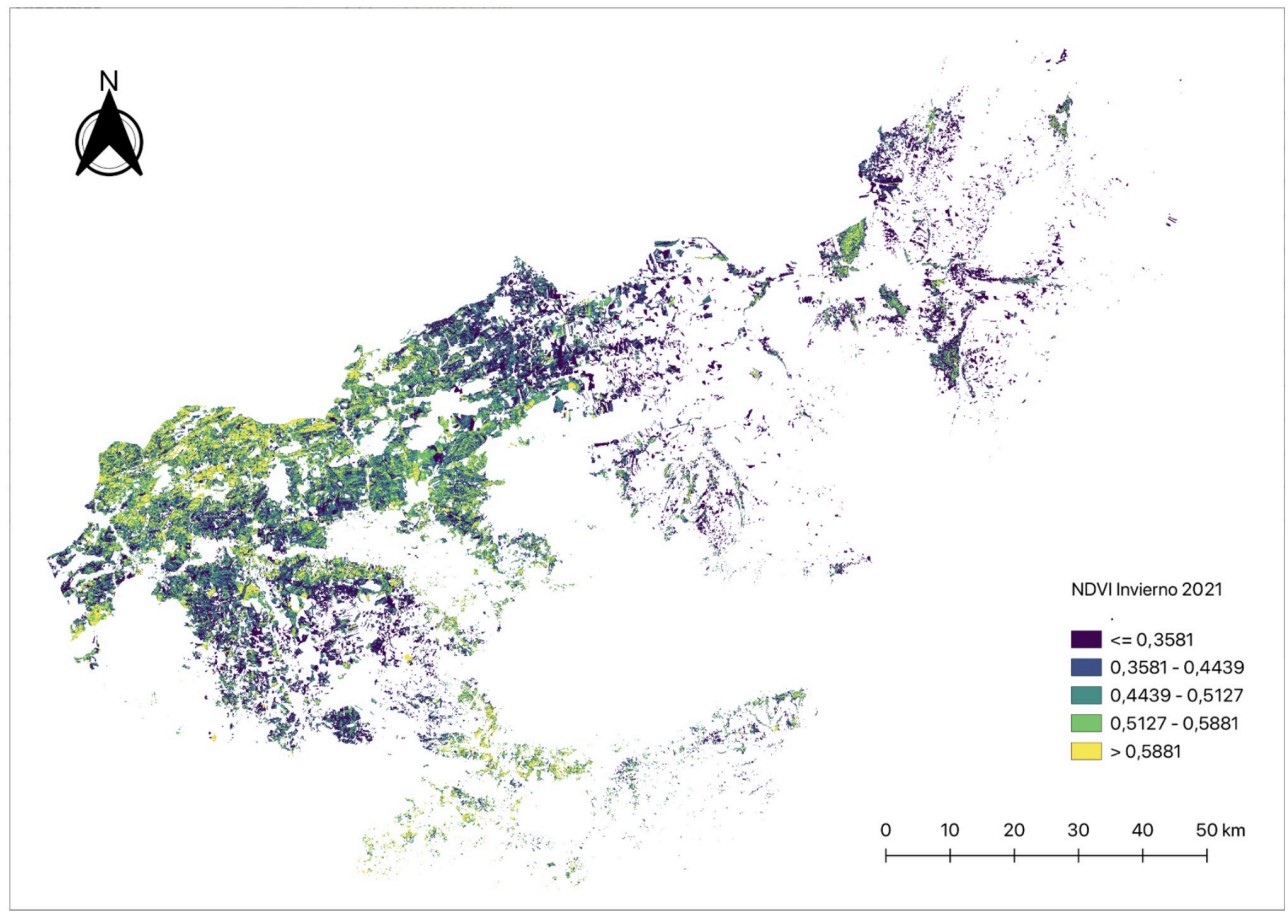

**Fig 3.**

All this information, as well as those related to weather variables, is obtained and processed by means of Google Earth Engine, a platform that is able to collect all this satellite imagery without the need of the user to download them, allowing remote processing. This platform combines an extensive catalog of satellite imagery and geospatial datasets with analysis capabilities at different scales. This approach enhances processing speed as data downloading is only required after final calculations in the form of numerical tables. Fig 3 shows the NDVI index calculated in the province of Jaen using Google Earth Engine.

The most pressing issue, given the division of seasons, is the presence of clouds, which disables the possibility of carrying out the calculations correctly. This problem is solved by increasing the number of images of the same area and combining them. This way the pixels that contain clouds can be eliminated and still are supplied by others which do have the information. The more images available, the more reliable the calculated indices are. However, the issue was the the absence of cloud-free images for certain municipalities during the winter of 2010 and 2011. Thus, the model operates properly from 2012 in order to make an accurate prediction.

As it is specified beforehand, several indexes are picked to be applied into this field of study, each one with specific characteristics and different utilities. This initial collection, as shown in Table 3, is drawn from scientific papers and various sources that focus on predicting crop

**Table 3. Main vegetation indices.**

| Index | Name |
|-------|------|
| NDVI | Normalized Difference Vegetation Index |
| EVI | Enhanced Vegetation Index |
| SAVI | Soil Adjusted Vegetation Index |
| CGI | Green Chlorophyll Index |
| ARVI | Atmospheric Leaf Index |
| RGR | Red Green Ratio |
| SIPI | Structure Insensitive Pigment Index |
| NBRI | Normalized Burn Ratio Index |
| NDRE | Normalized Difference Red Edge |
| GNDVI | Green Normalized Difference Index |
| MSAVI | Modified Soil Adjusted Vegetation Index |
| NDWI | Normalized Difference Water Index |
| MSI | Moisture Stress Index |
| NDMI | Normalized Difference Moisture Index |

outcomes and conducting surveys. For this work, it is preferable to have information about leaf chlorophyll, soil moisture in addition to vegetation stress and health.

The major drawback of satellite imagery is resolution, 30m in the case of Landsat 7 and 8. Typically, olive orchards do not have dense vegetation due to significant spacing between each plant. Therefore, it is important to consider that the area studied tends to have a predominance of soil exposure. After conducting an analysis of each index within the study area, and after performing various comparative assessments among them, certain indices are excluded from further consideration. The criteria for exclusion are based on their practical utility, the satellite sensor's capability to calculate them accurately, or the presence of redundant information. When feeding the predictive model, it is more efficient to provide simple, concise and precise information, avoiding to introduce noise to the model. The set of excluded indices are MSAVI and NDRE because of the lack of red-edge band in Landsat 7 and 8. EVI and GNDVI are not considered because of working mainly on dense vegetation. The RGR is rejected for being considered redundant, and finally NDWI and NBRI because of detecting water accumulation or fire respectively. Therefore, from Table 3 we finally consider the following indices in our study: ARVI, NDMI, GCI, NDVI, SAVI and SIPI, as referred in the following subsections.

**Weather information.** Weather information is obtained from different satellites. Data from meteorological stations is also available for temperature and precipitation, but covering the whole Jaen province implies inaccurate interpolation procedures.

The information obtained is grouped into seasons according to the olive tree necessities, and incorporated into the model forming labels rather than numerical ranges, which reduces the noise within the dataset. Furthermore, this approach normalizes all available information, creating a more straightforward system. Specific ranges are defined for both temperature and precipitation, generating labels such as 'low-medium-high'.

**Temperature.** This variable is monitored using MODIS constellation, given that it has spectral bands that allow the detection of surface temperature. Again, this is not the data with more resolution available, but is the most suitable one since it has information from 2009. MODIS provides two temperature values, one from the day where it is extracted the maximum, and another from the night where the minimum is selected. Regarding another satellites such as Landsat, this extreme ranges of MODIS makes it possible to consider temperature stress in trees.

Typically, the most relevant temperature data occurs in the warmer months such as May or June, as that is when the flower is forming. However, this article investigates early prediction, conducted in February, to forecast what will happen in the next harvest, making the temperature data from the previous year irrelevant for this study. Hence, the winter temperature is considered when making the prediction, as the potential temperatures of the following months are unknown.

Conversely, when considering precipitation data, calculations are necessary throughout the entire year. The total precipitation for each season is determined by the sum of the values, as opposed to calculating the average, as done with other variables.

**Rainfall.** For rainfall information, we use ECMWF data, based on Copernicus satellite images. This is the European Centre for Medium-Range Weather Forecasts, is an independent intergovernmental organization supported by many European nations. It specializes in producing and disseminating numerical weather predictions, which are crucial for meteorological services worldwide.

Specifically, ERAS5 (Earth Climate Reanalysis) project, which uses numerical models and observations to create a complete dataset on a global scale. It is not an image collection per se as the other cases, but is processed as one, obtaining data on millimeters of depth. As the data are processed by seasons that compile different numbers of months, the information obtained is the sum of rainfall in this time period.

## Modeling methodology

The methodology followed consists of the following phases, from the understanding of the data, its acquisition and preparation process, the generation of the models and, finally, the validation or analysis of the accuracy obtained. In supervised learning, when predicting numerical output variables, we are dealing with regression problems such as the case we present. There is no one-size-fits-all solution to every regression problem. There is no possibility of knowing which algorithm fits best in each case until it is tested. However, a pre-selection of algorithms can be made taking into account the characteristics of the input variables of our model as well as the type of output data. The output data of the model is twofold: the amount of olive harvest (the number of Kg of olives) and the amount of oil, which depends on the yield of the fruit. This information is collected for each campaign and for each one of all the province municipalities. In this approach, the study is based on a supervised machine learning analysis, where the value of an unknown variable (crop yield) is deduced from a few known variables.

The workflow of the methodology carried out consists of algorithms and data mining techniques used as follows in each phase:

1. *Data extraction and loading.* Harvest and olive oil datasets are public information provided by the regional governmental institution. Meteorological data is downloaded from publicly available web servers. Finally satellite images have been acquired and processed under Google Earth Engine. All this information is loaded into the database management system.

2. *Initial data preparation.* An exhaustive preliminary analysis of all the data is carried out using graphical representation techniques, such as histograms. This allows to examine the data thoroughly and filter out those whose dispersion or variability may generate inconsistencies in the study.

3. *Anomaly identification.* Anomaly detection is performed using a classification algorithm that has the ability to determine, with a certain probability, whether a data record fits the expected distribution pattern. The objective at this stage is to identify outliers in our data.

4. *Data transformation.* In this section, the input data is modified before being inserted into the model. The purpose is to adapt the formats, units, rescaling or sorting by ranges, among other actions, to optimize their use in the predictive models. This ensures that the models can make efficient data use. As an example, the specific transformations that have been made to the data are those that are typical of multi-source data integration. In the case of the meteorological data, file format conversions have been necessary, and in the case of the vegetation indices, a classification and subsequent labeling of the ranges has been required in order to be able to insert them into the models.

5. *Data aggregation.* The downloaded information is grouped by month; therefore, any other datasets that are added to the training data should be aggregated in the same way.

6. *Data integration.* In this analysis, a variety of data is available from different sources. In order to carry out the ML process, it is essential to unify all these data into a single source that serves as input for the predictive models. This requires the design of a suitable database that allows the input data to be properly stored in the system.

7. *Identification of how influential each input is on the target.* Before the model is generated, the level of influence of input data on the target attribute (Kg of olive fruit and olive oil) are analyzed in order to include or exclude them from predictive modeling.

8. *Regression algorithms applied.* Different regression algorithms are tested to perform the prediction. The objective of regression analysis is to determine the parameter values of a function that best fit an observational dataset. In addition to different families of regression functions, we also check different ways of measuring error.

The choice of crop prediction algorithms depends mainly on the ability to handle large and complex data sets with multiple input variables, which is our case. It also depends on the need to model linear or non-linear relationships and whether the types of data to be predicted are continuous or categorical. In addition, algorithms are required that are able to capture the interactions between the various agricultural factors that affect the yield of the olive crop, such as yield data from previous years, weather conditions and the spectral response of the crop as a function of soil properties, plant condition and agricultural practices. Our variable to be predicted, the crop yield, is a numerical and continuous data, so regression algorithms were applied, and within these, Random Forest, GLM, SVM with linear kernel and Gaussian were selected, and finally, tests with Neural Networks were also carried out. The reasons are that Random Forest, GLM, SVM with linear and Gaussian kernels, and Neural Networks share the ability to model complex nonlinear relationships through specific structures and transformations. Unlike regression algorithms such as Gradient Boosting Machines, which sequentially correct prior errors, or K-Nearest Neighbors, which depends on data proximity, these models can address a greater diversity of patterns in the data. In addition, they differ from Deep Learning techniques, which use multiple layers to extract progressively more abstract features and are more suitable for large volumes of data and unstructured features such as images and text.

**Workflow process.** The strategies employed are a series of mathematical operations that construct a design from the available information. To achieve this, a computational process first examines the data provided, detecting particular configurations of behavior or specific directions. Through multiple iterations, it uses the results of this evaluation to determine the ideal settings that will shape the design. These settings are then used on the full data set to identify actionable trends and provide detailed statistical analysis. This in-depth analysis of the data is carried out using an analysis module integrated into Oracle Data Mining software.

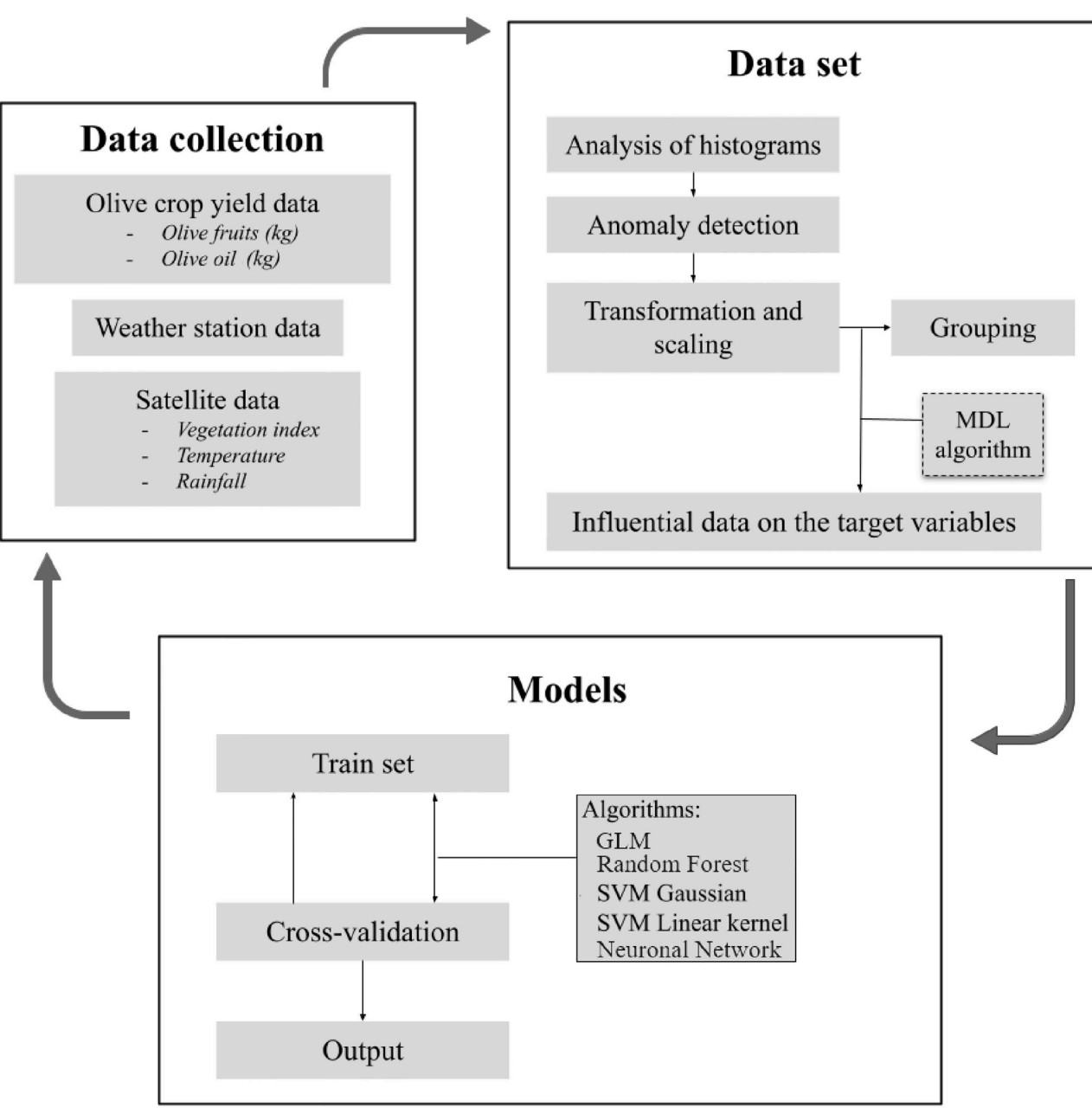

**Fig 4.**

Selecting the most appropriate algorithm for our work is a significant challenge. The algorithms used in each phase and the workflow are described in Fig 4.

Once the dataset is available, the following step is the detection of data anomalies. This is to identify unusual cases in apparently homogeneous data. Anomalies detection is essential to recognise fraudulent activities, unusual data and any other rare events that have the potential to cause major changes, often these are difficult to detect. A classification algorithm is used for this purpose because these anomalies can be considered as a particular case of classification. A classifier of a class develops a profile that generally describes a typical case in the training data.

Deviation from that profile is identified as an anomaly. Specifically, in this phase, the algorithm used has been the Support Vector Machine (SVM) algorithm, [49–51]. SVM is based on the fundamental idea of minimizing the hypersphere of the single class of examples in the training data, and consider all other examples outside the hypersphere as outliers or outside the distribution of the training data. This algorithm generates a prediction and a probability for each case in the score data. If the prediction is 1, the case is considered typical. If the prediction is 0, the case is considered anomalous. This behavior reflects the fact that the model is trained on data following a normal distribution. Also Generalized Linear Models (GLM) which is a pure linear regression algorithm [49, 52] is used, but it has been discarded because in early tests it performed worse than SVM.

The next phase consists of determining the level of influence of the variables used on the target attribute. In this case, the Minimum Description Length (MDL) algorithm [53] is used. This algorithm holds that, given a limited set of observed data, the best explanation is the one that allows the greatest compression of the data. MDL considers each attribute as a simple predictive model of the target class. Model selection refers to the process of comparing and ranking the simple predictive models. Running this algorithm produces a result in the range of −1 to 1. A value of 1 indicates that the features are highly related to the target, a value of 0 means that they are unrelated, and a negative value indicates that the feature is unrelated to the target, which could introduce noise into the study. In this analysis, only features with a weight greater than 0 are considered, discarding those with values equal to or less than 0.

Finally, regression analysis algorithms are used in the creation of the model, which focus on predicting the kilograms amount of olive crop and olive oil, both targets of this study. The choice to use regression is based on the fact that it is a data mining technique designed to anticipate numerical values over a continuous range. The regression process starts with a dataset in which the target values are already known. A regression algorithm calculates the desired value based on the predictors for each instance in that dataset. The relationships between the predictors and the target are encapsulated in a model that can then be applied to another dataset, where the target values are not known. In order to evaluate the effectiveness of regression models, calculations of various statistics are performed to assess the discrepancy between predictions and actual values. In a typical regression project, the historical dataset is divided into two parts: one to build the model and another to test the model. It is important to note that the year used for model testing is not part of the training dataset. As described in the State of the Art section, the type of target in this study makes regression algorithms the ideal ones to perform the regression, however, it is initially unknown which one will work best. To select the best one, it is necessary to perform preliminary tests and analyze the errors of each one. In this case, the preliminary tests completely ruled out the use of the GLM, since the resulting errors were not tolerable. The reason for such results is that it is a pure linear algorithm, prevailing nonlinear relationships between the variables of the dataset. Random Forest, SVM (with linear and Gaussian kernel) and neural networks were then applied. Although in their basic forms some of these algorithms inherently handle linear separations (such as linear SVM), all three can be extended or inherently possess the ability to model nonlinear relationships.

**Predictor variables selected.** Once the MDL algorithm is applied, the variables influencing both targets, quantity of Kg of olive crop and Kg of olive oil, are selected. These variables are shown in Table 4 calculated for each season and each year. As shown in the table, there are predictors whose values at various times in the previous, and also in the current year, are influential. This is the case, for example, for rainfall. In the crop prediction for the month of February (Winter), the values of rainfall both in the previous and current winter of the year of the prediction are considered to have an influence. The latter is often decisive for the crop yield, therefore its weight in the model is considered in this way.

**Table 4. Predictor variables considered for each municipality, and by year and season indicated.**

| Rainfall | Spring | Previous year |
|---|---|---|
| | Summer—Autumn | |
| | Winter | Previous year Actual year |
| Temperature | Winter | Actual year |
| Vegetation index | ARVI | Actual year |
| | NDMI | |
| | GCI | |
| | NDVI | |
| | SAVI | |
| | SIPI | |

The predictor variables must be categorized before applying the regression algorithms. For this purpose, the values are first normalized and then categorized by analyzing the histograms. The number of classes to be considered is set according to the variability of the data and then, a labeling process is applied.

The best predictors are those that best correlate the model. In this sense, numerous tests are carried out using different combinations of variables looking for the best set. By each predictors combination a model is obtained together with its corresponding Mean Absolute Error (MAE). This is calculated from each year's prediction, i.e. the difference between the predicted value and the real one. Afterwards, the total average of absolute errors of all the years is calculated, thus obtaining the consistency of the prediction for each set of predictors variables selected.

The MAE is the average of all absolute errors, and it is calculated as shown in Eq 1, where $n$ is the number of municipalities, $\Sigma$ is the summation symbol which means "add them all up", and $|xi - x|$ is the absolute error.

$$MAE = \frac{1}{n}\sum_{\xi=1}^{n}|x_\xi - x| \tag{1}$$

The RMSE, Eq 2, is also calculated for each model, because the two metrics are complementary. MAE measures average absolute errors, offering robustness against outliers and straightforward interpretability. However RMSE emphasizes larger errors due to its squaring step, making it sensitive to outliers and suitable for scenarios where large errors are critically undesirable. RMSE can disproportionately affect model evaluation due to its emphasis on larger errors.

$$RMSE = \sqrt{\frac{1}{n}\sum_{i=1}^{n}(x_\xi - x)^2} \tag{2}$$

**Accuracy evaluation.** After selecting the variables and ensuring all the information are accessible and standardized, the model is created. To gauge the model's effectiveness in predicting, we use the k-fold cross-validation technique [54]. This approach entails evaluating the model's predictive capability by isolating the data from a particular year and segregating it from the data used for model training. In the context of this research, it is utilized to validate the models' accuracy in predicting crop and oil production for a specific year. It's crucial to emphasize that the year in question has been deliberately omitted from the construction of

**Table 5. Error metrics of each regression algorithm using 2021 as control year.**

| Algorithm | RMSE (kg) | | Mean actual value (kg) | | Mean predictive value (kg) | |
|---|---|---|---|---|---|---|
| | Olive fruit crop | Olive oil | olive fruit crop | Olive oil | Olive fruit crop | Olive oil |
| Random Forest | 32945169,06 | 8022682,70 | 27741973,27 | 5757878,04 | 7515982,12 | 1134976,93 |
| SVM with lineal kernel | 11438152,91 | 1666390,32 | | | 24629414,86 | 5292237,67 |
| SVM with Gaussian kernel | 10959072,63 | 1280795,23 | | | 24733895,90 | 5243228,43 |
| Neuronal Netwok | 12013629,47 | 1938968,93 | | | 24153770,38 | 5217613,15 |

both models, as mentioned above. This study encompassed the period from 2009 to 2022. Specifically, we have constructed a predictive model using data spanning fourteen years, followed by an evaluation of its reliability using data from the tenth year. In the cross-validation process, the prediction for each year is checked. Therefore, several models are generated, one for each year to be checked. The year to be predicted is always excluded from the training data and compared with the actual data.

In this sense, and in order to make a first selection of algorithms, initial tests were carried out using Random Forest, SVM (with linear and Gaussian kernel) and Neural Networks algorithms. When evaluating the year 2021, the training data covers the years 2009 to 2020 and 2022, deliberately excluding the 2021 data used to evaluate the effectiveness of the model. In this first phase of algorithm selection, a specific set of predictors, the meteorological variables, was used. In a later phase, once the most suitable algorithms had been selected, an exhaustive analysis was carried out to select the most suitable predictors for the prediction model. In this preliminary analysis, the RMSE obtained by each algorithm for the year 2021 was analyzed, Table 5.

As explained above, in this initial phase of reporting results, the generated models are subjected to cross-validation. Specifically, a model is developed for each year to be evaluated. In the case shown in Table 5, the year 2021 is evaluated, so the deliberately 2021 is excluded from the dataset to evaluate the effectiveness of the model. Comparing the RMSE obtained for each algorithm, it can be confirmed that SVM models with two different kernels is the one that best fits the inherent characteristics of dataset with the target of our research, both olive fruit and oil crop yields.

## Results and Discussion

### Prediction using meteorological variables

The results obtained are highly dependent on the quality of the dataset. In this case, as the data come from different sources, it has been necessary to carry out an exhaustive review and a cleaning of outliers. This is essential to avoid introducing noise in the prediction and to be able to discern the most influential set of predictors from those that have less influence. It should also be taken into account that the same dataset can be used to predict different targets, with the weight of each predictor in the model differing according to the target considered.

A first test is carried out considering only meteorological predictors, those are, rainfall and temperatures. The result of the cross-validation, Fig 5, shows a mean absolute error in the prediction of olive crop quite similar to that obtained for olive oil.

In order to qualify the prediction obtained as good, the absolute error per municipality and per year is calculated. The total error of each test is obtained by adding up all the errors per year. In these first tests, high errors have been obtained. In general terms, it can be stated that the accuracy is low; however, it can be seen that using only rainfall gives slightly better results

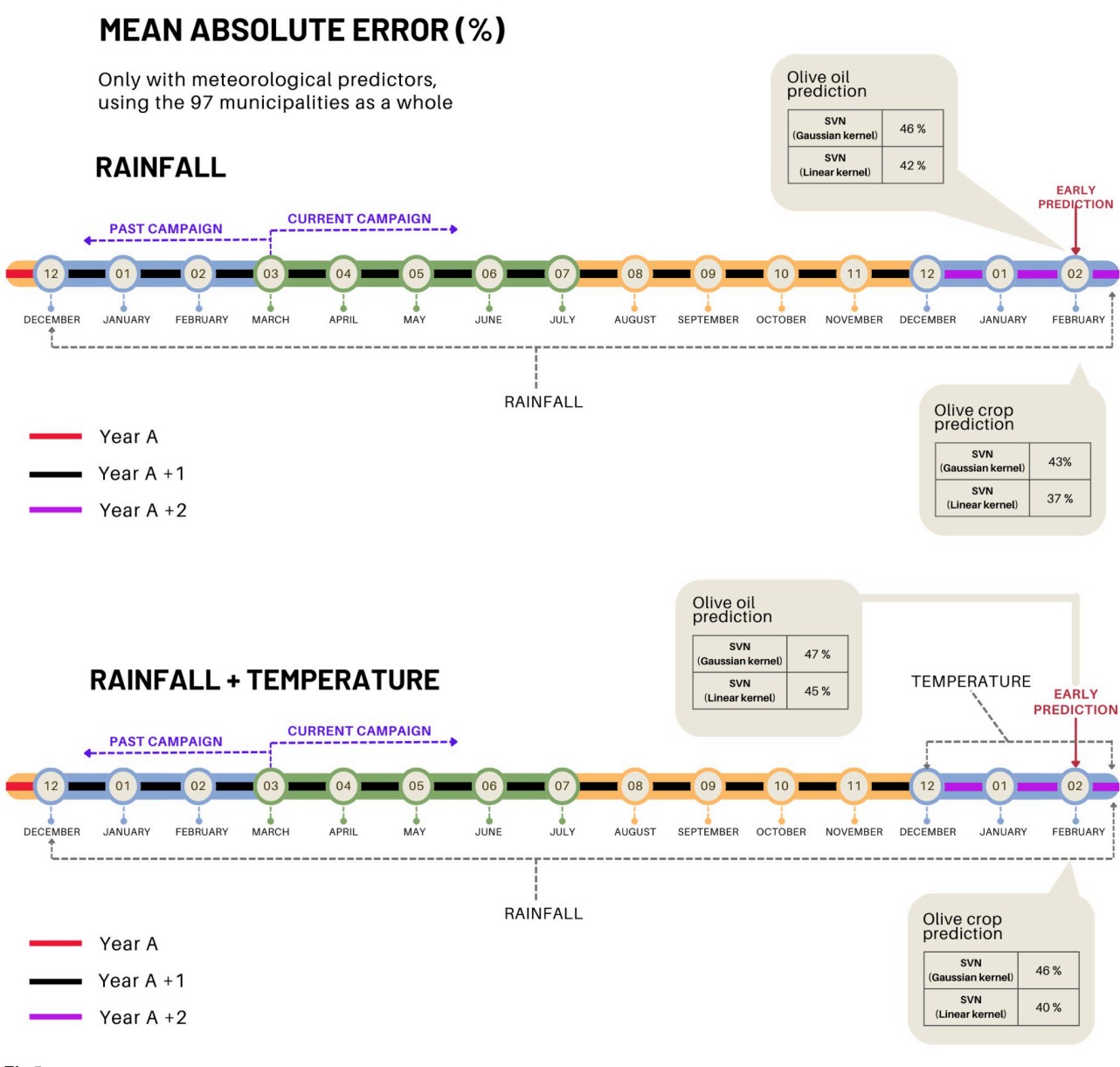

**Fig 5.**

than if it is combined with the temperature values. It seems that the temperature has little influence on the model according to the results observed in Fig 5. It can be seen that the temperature predictor not only does it not reduce the error of the model but slightly increases its value by introducing some noise. This may be due to the fact that temperature in the current winter is of little relevance to the yield of that year's crop.

Note that this result shows the average total error for the 97 municipalities considered. This error is also analyzed individually for each municipality. To do so, the MAE histogram is analyzed and it confirmed that a large number of municipalities have MAE values that are atypical with respect to the rest. These are a total of 19 municipalities. In this sense, and to guarantee the quality of the selection of predictors and the scope of our prediction, another test is carried

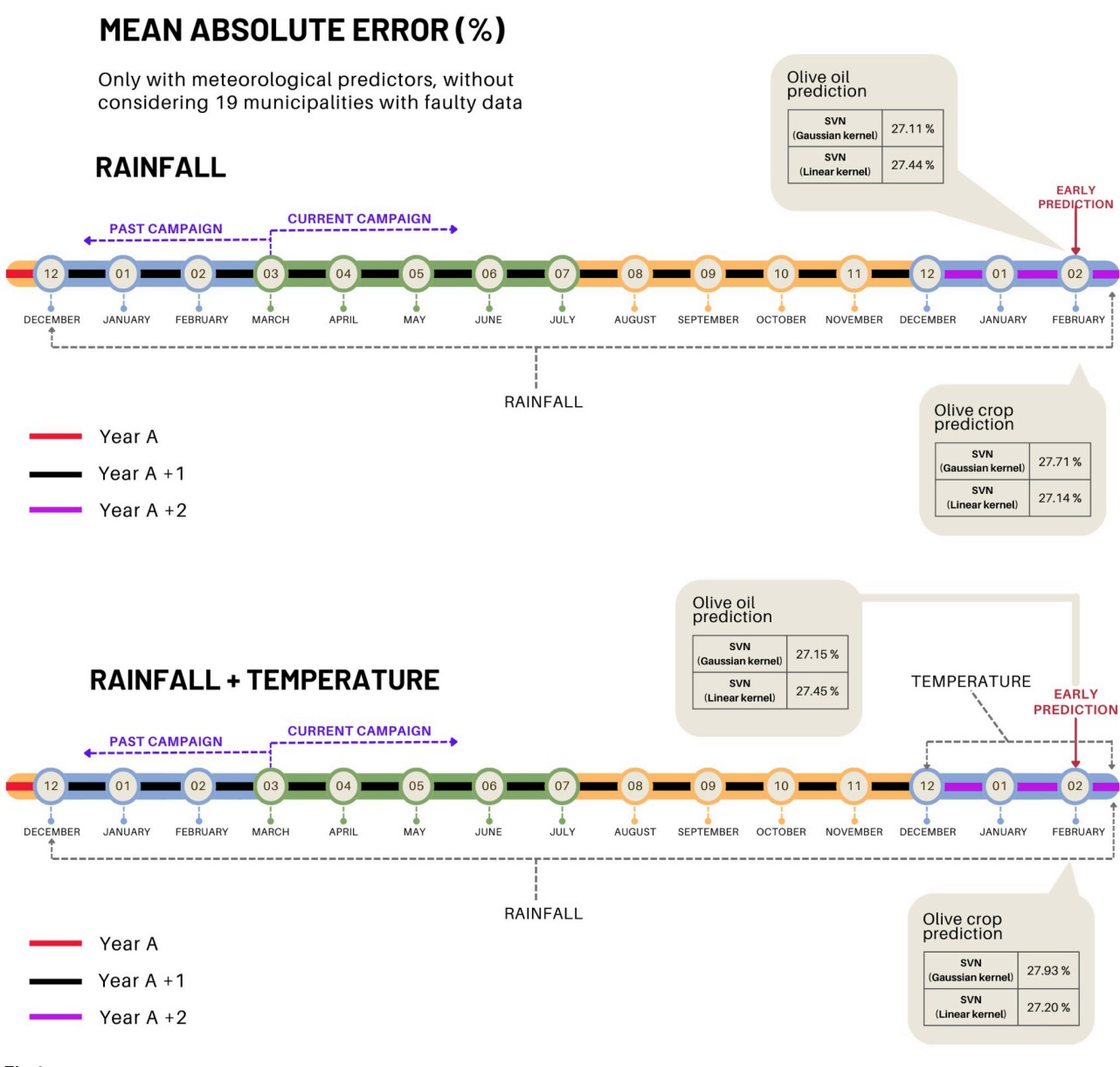

**Fig 6.**

out by discarding these municipalities from the dataset. The result is shown in Fig 5. Comparing Fig 6, it confirms the importance of proper data analysis and cleaning, as these 19 municipalities introduced significant noise into the model. Once these municipalities are removed from the dataset, the accuracy of the prediction improves considerably, around 27%.

Nevertheless, it is also observed that the SVM algorithm with Linear kernel, results in a better fitted model than using the Gaussian kernel. In this sense the default parameters were used by the system, Oracle Data Mining. They are shown in Table 6.

Overall, the low variability in the accuracy of the results using two different subsets of predictor variables may be due to the fact that these values as a whole differ little from one area to another. It may be necessary to use variables that better represent the state of olive orchards in

**Table 6. Parameters of the SVM algorithm with Gaussian kernel used to generate the predictive model.**

| SVM with Gaussian kernel | Parameters |
|---|---|
| Standard Deviation | 2,028.306 |
| Complexity Factor | 0.589921 |
| Kernel Function | Gausiano |
| Algorithm Name | Support Vector Machine |
| Active Optimisation | Enable |
| Automatic Preparation | Enable |
| SVM EPSILON | 0.040708 |
| Core Cache Size | 50,000.000 |
| Tolerance | 0.001 |

each season and in each area. It is therefore considered appropriate to introduce vegetation indices among the combinations of predictor variables.

## Prediction using vegetation index

The next test consists of introducing different vegetation indices into the set of predictors. In this case, after the thorough literature review described in previous sections, we have decided to consider those indices related to plant health and humidity values. In this sense, vegetation indices that provide values related to humidity can contribute with a better approximation of the state of the area after the weathering.

According to the literature review, a pre-selection of vegetation indices has been made. To determine the influence of each one on the model, the accuracy of each model is calculated by inserting rainfall values as predictors next to each index separately. The results are shown in Fig 7. The accuracy of the predictive model improves considerably by applying the Linear kernel, as the MAE is reduced to values of 26%. This accuracy in crop yield prediction, both for olives and olive oil is very useful for decision making. It should be borne in mind that this is an early prediction, before the start of the annual cycle of the olive tree and prior to the planning of investments to be made in tillage and phytosanitary products.

Once demonstrated the improvement of the predictive model by selecting the vegetation indices in the set of predictors, another test has been carried out using combinations of indices. Different combinations are made using ARVI (Atmospheric Resistant Vegetation Index) as a fixed index. The reason is that it minimizes the atmospheric dispersion effects caused by aerosols such as rain, fog, dust, smoke or air pollution. Therefore, it can be considered as a correction to the NDVI index. Thus the selected subsets are shown in Fig 8. In general terms, the improvement in prediction by integrating combinations of vegetation indices is not very significant, considering the effect on the region as a whole. A more detailed study, municipality by municipality, will probably allow to discern the most appropriate combination for each case (see Fig 9).

As can be seen in the figure, in general, the SVM algorithm with Gaussian kernel models the prediction better than the rest of the algorithms. However, when working on a larger scale, with a greater level of detail, the most appropriate algorithm for each municipality would have to be particularized since, as the detailed analysis shows, not all municipalities offer the same results.

Nevertheless, this study confirms for all cases the influence of the indices selected and the potential predictive improvement of combining them.

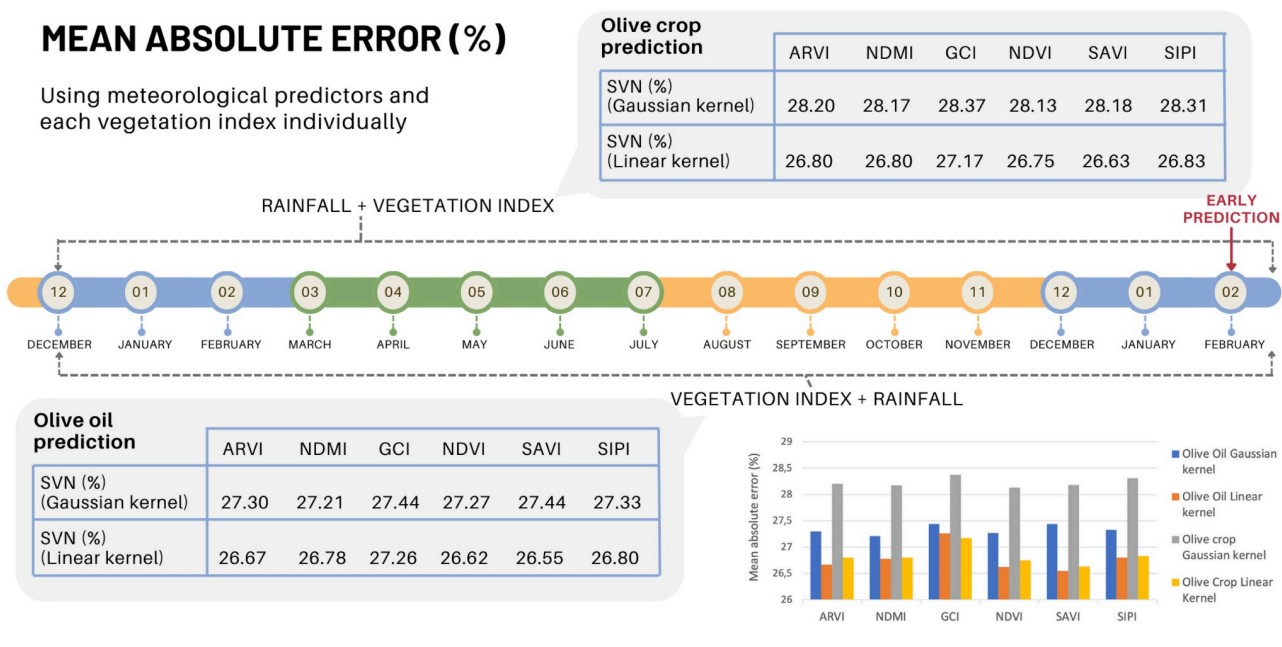

**Fig 7.**

## Conclusions

This study presents a workflow methodology describing the steps followed in the analysis of the predictive calculation of olive crop yield at an early stage. The correct selection of the predictor variables and the quality of these variables are fundamental. For this, the knowledge

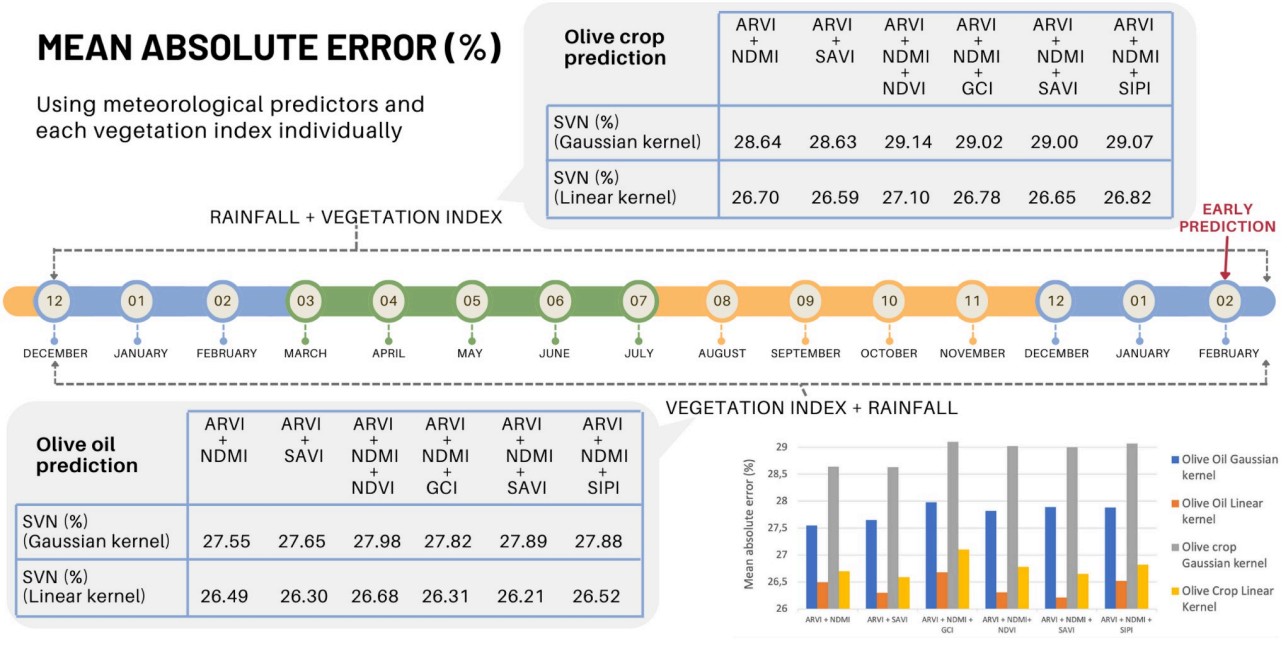

**Fig 8.**

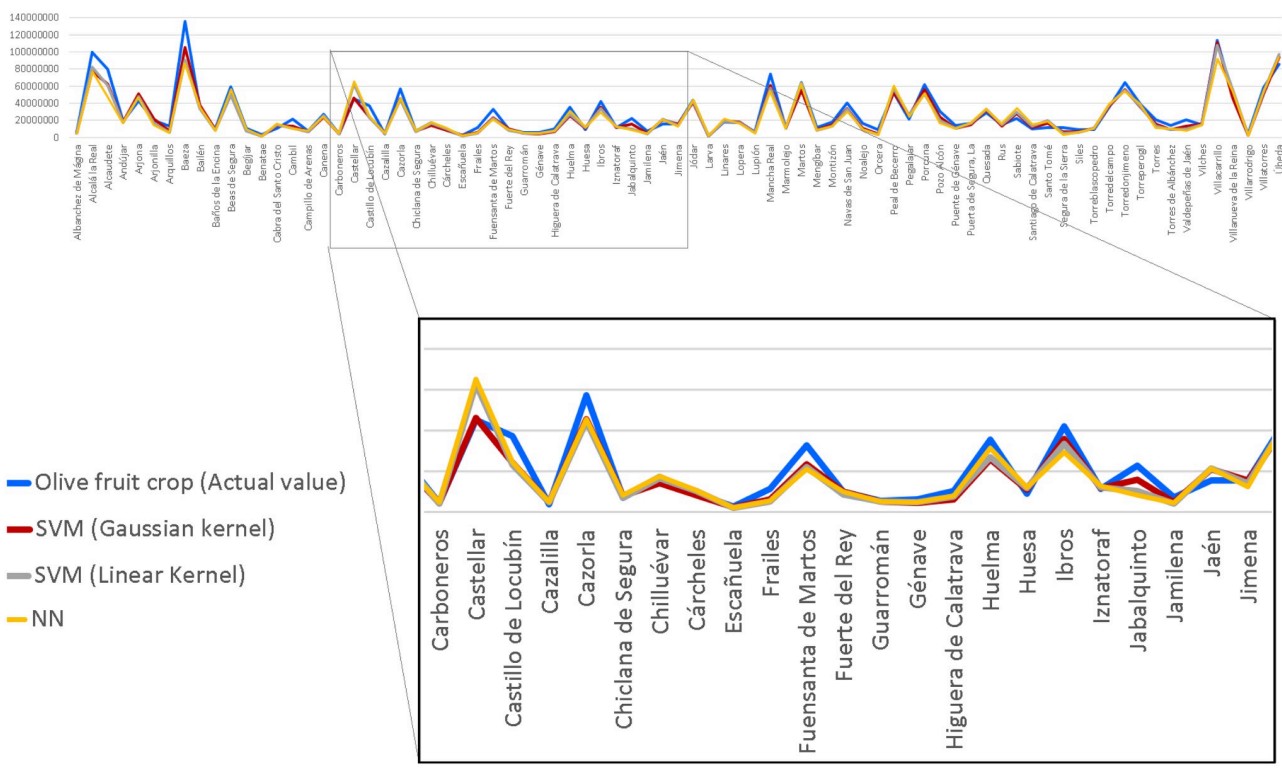

**Fig 9.**

about the type of crop and an exhaustive analysis of the influence of the variables considered guarantee to obtain an accurate predictive model. As has been shown, the availability of temporal data is essential. Only in this way a correct training of the regression model can be carried out.

In this research, predictive models of crop yield at an early stage of the olive growing season in Spain has been generated. This is before spring, just when the previous year's harvesting season has just ended and there is still no evidence in the field of how productive the next season will be. The targets of the predictive models are both olive fruit yield and oil yield. These are continuous numerical data, hence regression algorithms have been used when applying ML. It has been confirmed the SVM algorithm with Gaussian kernel provides the best predictive accuracy.

The integration of vegetation indices into the model improves crop yield prediction. This is due to the fact that a better diagnosis of the state of the plantation contributes to a good early prediction of its production. In this sense, satellite images have been fundamental in order to have sufficient temporality covering all the municipalities in the province of Jaen. The methodology developed is applicable to future works at different spatial scales, even at local or farm detail level. It is enough to have the historical harvest and meteorological data area under study, as well as UAV flights with multispectral sensors. The only limitation in this respect is the need to allow time to accumulate flights over the course of each year for as many years as possible.

## Author Contributions

**Conceptualization:** M. Isabel Ramos, Juan J. Cubillas.

**Data curation:** Ruth M. Córdoba.

**Formal analysis:** Juan J. Cubillas.

**Software:** Ruth M. Córdoba.

**Supervision:** Lidia M. Ortega.

**Writing – original draft:** M. Isabel Ramos, Ruth M. Córdoba, Lidia M. Ortega.

**Writing – review & editing:** M. Isabel Ramos, Lidia M. Ortega.

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
