## [Decision Letter · Decision Letter 0]

22 Feb 2024

PONE-D-24-04068Early season crop yield prediction using temporal data and satellite imagery with machine learning. A new avenue for planning in Spanish olive grovesPLOS ONE

Dear Dr. Ramos,

Thank you for submitting your manuscript to PLOS ONE. After careful consideration, we feel that it has merit but does not fully meet PLOS ONE’s publication criteria as it currently stands. Therefore, we invite you to submit a revised version of the manuscript that addresses the points raised during the review process.

We look forward to receiving your revised manuscript.

Kind regards,

Salim Heddam

Academic Editor

PLOS ONE

Journal Requirements:

3. We note that Figures 1 and 2 in your submission contain [map/satellite] images which may be copyrighted. All PLOS content is published under the Creative Commons Attribution License (CC BY 4.0), which means that the manuscript, images, and Supporting Information files will be freely available online, and any third party is permitted to access, download, copy, distribute, and use these materials in any way, even commercially, with proper attribution. For these reasons, we cannot publish previously copyrighted maps or satellite images created using proprietary data, such as Google software (Google Maps, Street View, and Earth). For more information, see our copyright guidelines: http://journals.plos.org/plosone/s/licenses-and-copyright.

a. You may seek permission from the original copyright holder of Figures 1 and 2 to publish the content specifically under the CC BY 4.0 license.  

Additional Editor Comments:

Reviewer 1#:

The topic seems interesting and it seems that the authors have done a good job. However, the paper is very lengthy and unnecessary details are added which makes the paper boring and distracting. The extra details may be good for inclusion in thesis but not in a research paper.

Further, the model does not seem to achieve a very good performance. 26% error seems to be too high. I will recommend the authors to try some other models such as ANN, Random Forest, or even considering deep learning models is not a bad choice.

Some other point that needs consideration are;

1. The title requires reconsideration. Why " A new avenue for planning in Spanish olive groves" is important.

2. the first few paragraphs of Section 2 is not about the state of the art but can be a good motivation for the work. Please reduce the length of the text and focus on the relevant works. I will also recommend removing some irrelevant works and add some recent works instead.

3. The overall language quality is reasonable, however, at some points, it requires proof read. For instance Line 161 "... on Feb ..." => "... in Feb...". There are other such instances as well.

4. Line 331, "This information is extracted from .... " what information are you referring to?

Recommendation:

I like the topic and the amount and diversity of data that is being collected. However, the paper in current form and with the current models is not interesting. I will recommend a complete re-write, making it crisp and to the point. Most importantly, I will recommend implementation more ML models such as ANN, random forest and deep learning models.

For guidelines, I found the following papers as good examples to follow;

1. doi.org/10.1002/agj2.21382

2. doi.org/10.1016/j.eswa.2021.115511

3. doi.org/10.1016/j.rse.2018.11.032

Some of these works may not be regression based problems, but it give a good idea about writing such a paper.

Reviewer 2#:

The state of the art and conclusions should discuss the environmental and socio-economic challenges of olive production as a socio-ecological system. Olive production data from Spain should be compared with other regions of the world as well as their problems, challenges and alternatives (a SWOT analysis may be useful).

The authors do not discuss the implications of their research for sustainable development.

The MS is very extensive. The authors could reduce some sections such as the state of the art, spectral indices among others and strengthen the discussion and conclusions.

Reviewer 3#:

Overall, the authors have conducted an excellent study to examine prediction analysis for olive trees with multiple dataset. However, the following information is not clear in the publication. Therefore, the paper needs improvement and should be written in a more understandable manner.

1. The introduction section should be expanded within the scope of methodology and prediction analyses.

2. In the final paragraph of the introduction, a brief overview should be provided on how the low spatial resolution of satellite images used in the classification of a single tree or tree clusters was addressed in the study.

3. The Table 4 should be within the content of section 3.2.3.

4. The visual representation of processed and utilized satellite image data with the Google Earth Engine system should be provided.

5. In “Data Transformation” section, It should be written which data formats, scales, etc. have been modified

6. Tables 10 and 11 should be explained in more detail in “4.2. Prediction using vegetation index”.

7. Why is prediction not made solely with vegetation indices apart from meteorological data? If feasible, it would be beneficial to make a prediction solely based on vegetation indices and share the outcome.

8. It is understood from the publication that prediction error values have been determined. However, upon reviewing the introduction and publication title, it is implied that the quantity of the crop to be harvested for the studied area will be determined. To address this complexity, explanations need to be provided within the text.

Reviewer 4#:

This work demonstrates the potential for advanced technologies and methodologies to significantly benefit agricultural management and decision-making processes. The methodological rigour explored a d the resulting prediction accuracy is worth noting.

Reviewer 5#:

1- The paper title is too long, so it need to be revised.

2- The abstract is not highlighted the paper findings well and need to be rewritten.

3- Authors did not mention the official sources and agencies from where they obtained the dataset.

4- Absolute error is a measure of how far off' a measurement is from a true value or an indication of the uncertainty in a measurement. So 26% means authors are very far off from the true value.

5- The paper novelty is low, all models and tools used are exist in the state-of-the-art.

6- To reflect the robustness of author proposed method, a comparison with two or more state of art models is preferred.

7- I doubt it can have said to be a journal paper with significant contributions.

Reviewer 6#:

Paper is welln written but methodology section can be improved by providing figure of methodology working to improve the understanding.

Improve the English and reduce jorgans to.improve reader understanding.

Improve the conclusion section of the paper by more clearing the expected outcomes received from this research

Reviewer 7#:

Review comments

This work mainly employed SVM to predict the olive crop and oil yield using satellite data. However, there are significant issues throughout this manuscript thus leading it is unsound at current version.

Significantly, this work fails to offer compelling evidence of the success of the main prediction task, which is using an SVM regression model for crop yield prediction. It only presents Mean Absolute Error (MAE), which is insufficient for evaluating the model's performance effectively (or cannot fully capture the severity of prediction errors) within the specific context of crop yield prediction. Additionally, several steps in machine learning lack clarity without further elaboration even the methods section hints at some clues. Specifically, normalization or standardization, anomaly detection, hyperparameter tuning, and feature selection require thorough calibration and detailed explanation.

Thus, I cannot currently endorse this manuscript for publication in this journal as it requires substantial improvements to meet the promised standards in academic research.

My detailed review comments are listed below.

1. Introduction

1.1 The title needs revision: there's no need to incorporate ‘temporal data’, as the result in this manuscript has not addressed the importance of including temporal trends through any method involving modelling in the time dimension (e.g., ARIMA, RNN, LSTM). In terms of the subtitle, it is preferable to phrase it as follows: ‘A case study of olive groves in Jaden Span’ as the insufficient discussion or analysis regarding the planning implications in this manuscript.

1.2 The research gap and the research question should be clearly addressed in introduction section.

1.3 The early crop prediction requires a clearer and more detailed definition following a background overview of the olive growing season provided in this section. The early crop prediction refers to forecasting tasks conducted in the early stages of the season, typically around February (in the case of olive crop), utilizing data from previous years? Also, a discrimination of early prediction and late prediction can be included in this section.

1.4 In line 67, a typo: ‘this q work’

2. Related work (State of art)

2.1 This section requires further revision as it encompasses multiple aspects. First, the description of the Spanish olive case, including the importance of early prediction of olive yield, could be incorporated into the case study section (or study area). Second, the review of key factors in crop yield prediction needs to be organized to list variables and their related indices more succinctly. Integrating this information would enhance the informativeness of Table 2.

2.2 In line 178, a typo: ‘stydy’

2.3 Creating a table for three numbers is unnecessary (Table 1)

3. Data and method

3.1 In line 346, ‘in which only those plots of olive trees have been considered’, Does this imply that several cities (municipalities) are selected from all 97 municipalities here? Please provide clarification and mapping details.

3.2 In line 347, it mentioned ‘process the spectral images’, what kind of information has been collected from Google Earth Engine? The polygons in each city or others?

3.3 In line 349, what is simplification process? More details should be provided regarding this geoprocessing (remove line strings in a defined polygon in Figure 2 ?).

3.4 For line 360 to 372, why is the olive growing cycle discussed in the dataset instead of incorporating relevant literature to establish a connection with the study's objective, such as early and late prediction?

3.5 In subsection 3.2.2 Weather information, why is the weather variable, sourced from satellite data, treated separately from the satellite data discussed in section 3.2.3 ? Any specific intention? Furthermore, please explain how the three labels mentioned from line 435 to 443 are derived based on temperature and precipitation. Another concern is the presentation of Temperature and Rainfall data in 3.2.3, which requires clarification (different information or other intentions).

3.6 In section 3.2.3 satellite data, the focus should be more on the indices utilized rather than on not using them in this study. Therefore, consolidating all indices and their respective data sources into one table can enhance clarity in this section.

3.7 In table 6, the MSI is not included in Table 5.

3.8 In line 528, what is ECMWF (have not found any clues through the whole manuscript)?

4. Model

4.1 In Initial data preparation, any results or any presentation for each distribution of the variable (in result section)? Or can a statistical description of all independent variables and dependent variables be provided in a Table?

4.2 In line 567, what kind of algorithm was used in anomaly identification?

4.3 In data transformation, does it refer to the standardization or normalization of the data? If so, what specific technique is being referred to? Please provide the equation for the technique mentioned.

4.4 In line 581, what is the ‘unify all these data into a single source’?

4.5 Does the point 7 mean the feature selection? What is the meaning of ‘we also check different ways of measuring error’? using different indices?

4.6 The 8 points can be aggregated to workflow process subsection by providing more details of model/algorithms used in each step. Section 3.3 can be organized flowing the workflow processing orders.

4.7 From line 609 to 620, please clarify how SVM has been applied to AD. If so, is it employed for classification, and if it is, is it binary or multi-class classification? Additionally, which type of outliers will be eliminated in this context? Alternatively, is this technique referring to unsupervised one-class SVM? Another concern is the intention of SVM used for AD in this dataset? It also indicates that the performance of GLM is worse compared to SVM; to what degree is this disparity?

4.8 What kind of index (why the value of feature above 1 is selected) is used and how the MDL help to select the features? In Line 630, it mentioned model selection which need more specification.

4.9 What is the meaning of ‘the restrictions only depend on the limitations imposed by the hardware used’ in line 659?

4.10 In line 671, what is the intention of categorization? Does this mean that it transitions from a regression task to a classification task?

4.11 From line 680 to 683, it should be more specific while you use the MAE, the n is the sample numbers, so if the prediction for one year is based on 97 cities (the geospatial unit of analysis should also be mentioned in the study area), so the n is 97. If you implement 10 times prediction for 10 years, you will get 10 MAEs here.

4.12 In line 690, what is the value of k?

4.13 The hyperparameter processing SVM is missing, i.e., selection the optimised parameters?

4.14 The olive oil is litres in Figure 3 but Kg in other places in the manuscript.

4.15 A concern is that what is the mathematic representation of VI index for a city-level as an input in the SVM model? A vector or a matrix? Please make more specific for each variable input and the sample numbers (i.e., cities and annual months) in the beginning of the workflow introduction.

5. Results and discussion

5.1 It is preferable to create a plot depicting the points based on predicted values and actual values. This approach facilitates the generation of metrics such as R-square (RMSE and MSE can be considered as well). Furthermore, why MAE is solely showed in the training set and not the testing set as it mentioned trained using data from seven years to predict outcomes for another year in line 697.

Another concern of MAE is the type of value utilized for calculation, whether it's the normalized value (ranging from -1 to 1) or the actual value (potentially 1000,000 Kg of olive crop). Demonstrating the error in actual values is essential for assessing the efficiency and accuracy of the SVM in predicting the olive crop and oil.

5.2 It is confused that line 715 involves predicting the year 2017 using data from 2018 to 2021. Could this be one part of the cross-validation procedure? There is a need for clarification regarding the training (including hyperparameters, cross-validation) and testing processes (alongside which part of data) involved in this case.

5.3 The discussion is brief, particularly regarding the limitations identified in this study, such as issues with the model and data representation.

Reviewer 8#:

The aim of the study was to predict accurate yield data for olive groves in the province of Jaen in the Andalusia region of southern Spain at the beginning of the growing season using machine learning algorithms and a thorough analysis of the predictor variables.

The outline of the paper is generally appropriate and has some strengths, although some major details need to be revised before acceptance:

1. The introduction (section 1) lacks proper referencing, making it essential to include references in the sentences to support the content effectively.

2. The second part deals well with the subject of theory and also the review of related studies, in this sense this part was well written.

3. The third section of the article elucidates the study area, providing comprehensive details on data collection and corresponding analysis, presented in a well-crafted manner.

4. The section on the results is also acceptable, as it clearly presents the results in two subsections - prediction based on meteorological variables and prediction based on the vegetation index.

5. Nevertheless, it is essential to present the discussion section clearly in order to enable a thorough examination of the results and their comparison with the findings from other studies. In this way, the unique contributions of the authors in the context of this study should be emphasised.

6. In addition, as can be seen in the conclusion section, the strengthening of the content in relation to the research implication as well as its limitations and suggestions for future studies should be clearly presented.

Reviewer 9#:

The authors have addressed all the questions and comments, and this manuscript is good to publish. The statistical analysis has been performed well according to the data obtained. The English Language and grammar meet the requirements to get published in Plos One. This research work will be helpful for better understanding of the early season crop yield prediction with machine learning as it is demanding in the industry and its a new approach in this field.

Reviewer 10#:

The study may focus on the machine learning section

In general, the article should be reviewed in accordance with the journal writing rules.

table 6 needs to be reorganized

Results section should be expanded.

Reviewer 11#:

I appreciate the authors' effort in applying SVM to predict crop yields using temporal data. However, there are several limitations and areas of improvement that I'd like to highlight:

1. A more detailed comparison with traditional time series analysis methods is necessary. It would be beneficial to include a discussion on the specific advantages of SVM in handling the nonlinearities and complexities of the data involved in this study.

2. The results section primarily relies on point estimates for performance metrics. In real-world scenarios, understanding the uncertainty around these estimates is crucial. The authors should consider providing confidence intervals.

3. For any machine learning study, it's crucial to compare the performance against standard baselines.

4. This work doesn't thoroughly justify the choice of the SVM model for this particular task. Are there specific characteristics of the dataset or problem domain that make SVM particularly suited, or was it an arbitrary choice?

5. SVM has several hyperparameters, such as the regularization constant (C), the kernel type (linear, polynomial, radial basis function, etc.), and parameters specific to those kernels (like degree for polynomial or gamma for RBF). These parameters can significantly affect performance. Not discussing them leaves out crucial aspects of the model's behavior. Also, without knowledge of the chosen hyperparameters, there's the concern of potential overfitting. For example, an SVM with a small value of C might fit the training data very closely, possibly capturing noise and leading to poor generalization on unseen data. To strengthen the paper, the authors should have included details on the SVM parameters, the rationale behind their selection, and potentially even results from a hyperparameter tuning process. This would provide more clarity and confidence in the presented results.

Reviewer 12#:

1-The abstract must clearly articulate the originality of the investigation.

2-The authors did not specify the method used to determine the sample size.

3-If the authors draw A flowchart to the workflow in the methodology, the part will help link the audiences to how this method advances current knowledge and will help the audiences follow the results quickly.

4-The researchers need to mention the experimental design they used in this manuscript to help the readers follow the results and the conclusions.

Reviewer 13#:

1. Based on the detailed examination of the “Materials and Methods” section and subsequent results and discussion, the manuscript demonstrates a methodologically sound approach to predicting olive crop yields using satellite imagery and machine learning techniques, specifically SVM with Linear and Gaussian kernels. The study incorporates rigorous data preprocessing, anomaly detection, and transformation steps to optimize the input for predictive modeling. It also employs regression and analysis and evaluates the influence of various predictors on the target variables (kg of live fruit and olive oil).

The use of k-fold cross-validation for model accuracy assessment and the detailed analysis of mean absolute error (MAE) across different models and predictor sets further illustrate the rigorous statistical and experimental methods applied. The results section substantiates the effectiveness of integrating meteorological variables and vegetation indices into the predictive model, highlighting improvements in prediction accuracy through these methods.

Given these observations the manuscript can be considered partly technically sound as it follows rigorous experimental procedures, employs appropriate controls (e.g., cross-validation), and uses relevant sample sizes for the machine learning model development and validation process. The conclusions drawn from the data presented are supported by the analysis conducted, indicating that the study’s outcomes are rooted in the data obtained and processed through the described methodologies.

However, the assessment of “partly” also acknowledges that while the manuscript demonstrates technical rigor in its approach, the effectiveness and robustness of the predictive models could benefit from further validation across broader datasets or independent validation sets to strengthen the conclusions drawn. Additionally, discussions on the limitations of the study, potential biases in the data or model, and the generalizability of the findings to other regions or crop types would enhance the manuscript’s depth and scientific robustness.

2. Based on the review of the "Materials and Methods" section and related content, including the statistical methods applied for data analysis and model evaluation, the statistical analysis in the manuscript has been performed appropriately and rigorously.

3. Based on the information provided in the “Materials and Methods” and other relevant sections of the manuscript, the authors have indicated that the primary data sources for their study are publicly accessible, including datasets form regional governmental institutions and publicly available web servers for meteorological data. The use of Google Earth Engine for processing satellite images is also mentioned, suggesting the data used in the study are accessible through public platforms or repositories. However, the manuscript does not provide explicit details on accessing these datasets, such as direct links, access instructions, or deposit information in a public repository specific to the study’s datasets.

Given the lack of explicit instructions or links for direct data access or the deposit of processed data in a recognized public repository, and in the absence of a clearly stated Data Availability Statement that meets the PLOS Data policy’s requirements for unrestricted access to all underlying data, the answer would be “No”. The manuscript does not fully comply with the requirements to make all data underlying the findings fully available without restrictions, as per the PLOS Data policy guidelines.

4. Based on the sections of the manuscript reviewed, including the abstract, introduction, materials and methods, results and conclusions, the manuscript is presented in an intelligible fashion and is written in standard English. The test maintains a formal and scientific tone, appropriate for an academic audience, with technical terms and concepts explained in a manner that should be accessible to readers familiar with the field. There are no significant typographical or grammatical errors noted in the sections reviewed, and the manuscript follows a logical structure that facilitates understanding of the study’s objectives, methods, results and implications.

Reviewers' comments:

Reviewer's Responses to Questions

**Comments to the Author**

1. Is the manuscript technically sound, and do the data support the conclusions?

Reviewer #1: Partly

Reviewer #2: Yes

Reviewer #3: Partly

Reviewer #4: Yes

Reviewer #5: Partly

Reviewer #6: Yes

Reviewer #7: Partly

Reviewer #8: Partly

Reviewer #9: Yes

Reviewer #10: Yes

Reviewer #11: Partly

Reviewer #12: Yes

Reviewer #13: Partly

2. Has the statistical analysis been performed appropriately and rigorously? 

Reviewer #1: N/A

Reviewer #2: Yes

Reviewer #3: Yes

Reviewer #4: Yes

Reviewer #5: Yes

Reviewer #6: Yes

Reviewer #7: No

Reviewer #8: Yes

Reviewer #9: Yes

Reviewer #10: N/A

Reviewer #11: No

Reviewer #12: Yes

Reviewer #13: Yes

3. Have the authors made all data underlying the findings in their manuscript fully available?

Reviewer #1: No

Reviewer #2: No

Reviewer #3: Yes

Reviewer #4: Yes

Reviewer #5: No

Reviewer #6: Yes

Reviewer #7: No

Reviewer #8: Yes

Reviewer #9: Yes

Reviewer #10: Yes

Reviewer #11: No

Reviewer #12: Yes

Reviewer #13: No

4. Is the manuscript presented in an intelligible fashion and written in standard English?

Reviewer #1: No

Reviewer #2: Yes

Reviewer #3: Yes

Reviewer #4: Yes

Reviewer #5: Yes

Reviewer #6: Yes

Reviewer #7: No

Reviewer #8: Yes

Reviewer #9: Yes

Reviewer #10: Yes

Reviewer #11: Yes

Reviewer #12: Yes

Reviewer #13: Yes

5. Review Comments to the Author

Reviewer #1: The topic seems interesting and it seems that the authors have done a good job. However, the paper is very lengthy and unnecessary details are added which makes the paper boring and distracting. The extra details may be good for inclusion in thesis but not in a research paper.

Further, the model does not seem to achieve a very good performance. 26% error seems to be too high. I will recommend the authors to try some other models such as ANN, Random Forest, or even considering deep learning models is not a bad choice.

Some other point that needs consideration are;

1. The title requires reconsideration. Why " A new avenue for planning in Spanish olive groves" is important.

2. the first few paragraphs of Section 2 is not about the state of the art but can be a good motivation for the work. Please reduce the length of the text and focus on the relevant works. I will also recommend removing some irrelevant works and add some recent works instead.

3. The overall language quality is reasonable, however, at some points, it requires proof read. For instance Line 161 "... on Feb ..." => "... in Feb...". There are other such instances as well.

4. Line 331, "This information is extracted from .... " what information are you referring to?

Recommendation:

I like the topic and the amount and diversity of data that is being collected. However, the paper in current form and with the current models is not interesting. I will recommend a complete re-write, making it crisp and to the point. Most importantly, I will recommend implementation more ML models such as ANN, random forest and deep learning models.

For guidelines, I found the following papers as good examples to follow;

1. doi.org/10.1002/agj2.21382

2. doi.org/10.1016/j.eswa.2021.115511

3. doi.org/10.1016/j.rse.2018.11.032

Some of these works may not be regression based problems, but it give a good idea about writing such a paper.

Reviewer #2: The state of the art and conclusions should discuss the environmental and socio-economic challenges of olive production as a socio-ecological system. Olive production data from Spain should be compared with other regions of the world as well as their problems, challenges and alternatives (a SWOT analysis may be useful).

The authors do not discuss the implications of their research for sustainable development.

The MS is very extensive. The authors could reduce some sections such as the state of the art, spectral indices among others and strengthen the discussion and conclusions.

Reviewer #3: Overall, the authors have conducted an excellent study to examine prediction analysis for olive trees with multiple dataset. However, the following information is not clear in the publication. Therefore, the paper needs improvement and should be written in a more understandable manner.

1. The introduction section should be expanded within the scope of methodology and prediction analyses.

2. In the final paragraph of the introduction, a brief overview should be provided on how the low spatial resolution of satellite images used in the classification of a single tree or tree clusters was addressed in the study.

3. The Table 4 should be within the content of section 3.2.3.

4. The visual representation of processed and utilized satellite image data with the Google Earth Engine system should be provided.

5. In “Data Transformation” section, It should be written which data formats, scales, etc. have been modified

6. Tables 10 and 11 should be explained in more detail in “4.2. Prediction using vegetation index”.

7. Why is prediction not made solely with vegetation indices apart from meteorological data? If feasible, it would be beneficial to make a prediction solely based on vegetation indices and share the outcome.

8. It is understood from the publication that prediction error values have been determined. However, upon reviewing the introduction and publication title, it is implied that the quantity of the crop to be harvested for the studied area will be determined. To address this complexity, explanations need to be provided within the text.

Reviewer #4: This work demonstrates the potential for advanced technologies and methodologies to significantly benefit agricultural management and decision-making processes. The methodological rigour explored a d the resulting prediction accuracy is worth noting.

Reviewer #5: 1- The paper title is too long, so it need to be revised.

2- The abstract is not highlighted the paper findings well and need to be rewritten.

3- Authors did not mention the official sources and agencies from where they obtained the dataset.

4- Absolute error is a measure of how far off' a measurement is from a true value or an indication of the uncertainty in a measurement. So 26% means authors are very far off from the true value.

5- The paper novelty is low, all models and tools used are exist in the state-of-the-art.

6- To reflect the robustness of author proposed method, a comparison with two or more state of art models is preferred.

7- I doubt it can have said to be a journal paper with significant contributions.

Reviewer #6: Paper is welln written but methodology section can be improved by providing figure of methodology working to improve the understanding.

Improve the English and reduce jorgans to.improve reader understanding.

Improve the conclusion section of the paper by more clearing the expected outcomes received from this research

Reviewer #7: Review comments

This work mainly employed SVM to predict the olive crop and oil yield using satellite data. However, there are significant issues throughout this manuscript thus leading it is unsound at current version.

Significantly, this work fails to offer compelling evidence of the success of the main prediction task, which is using an SVM regression model for crop yield prediction. It only presents Mean Absolute Error (MAE), which is insufficient for evaluating the model's performance effectively (or cannot fully capture the severity of prediction errors) within the specific context of crop yield prediction. Additionally, several steps in machine learning lack clarity without further elaboration even the methods section hints at some clues. Specifically, normalization or standardization, anomaly detection, hyperparameter tuning, and feature selection require thorough calibration and detailed explanation.

Thus, I cannot currently endorse this manuscript for publication in this journal as it requires substantial improvements to meet the promised standards in academic research.

My detailed review comments are listed below.

1. Introduction

1.1 The title needs revision: there's no need to incorporate ‘temporal data’, as the result in this manuscript has not addressed the importance of including temporal trends through any method involving modelling in the time dimension (e.g., ARIMA, RNN, LSTM). In terms of the subtitle, it is preferable to phrase it as follows: ‘A case study of olive groves in Jaden Span’ as the insufficient discussion or analysis regarding the planning implications in this manuscript.

1.2 The research gap and the research question should be clearly addressed in introduction section.

1.3 The early crop prediction requires a clearer and more detailed definition following a background overview of the olive growing season provided in this section. The early crop prediction refers to forecasting tasks conducted in the early stages of the season, typically around February (in the case of olive crop), utilizing data from previous years? Also, a discrimination of early prediction and late prediction can be included in this section.

1.4 In line 67, a typo: ‘this q work’

2. Related work (State of art)

2.1 This section requires further revision as it encompasses multiple aspects. First, the description of the Spanish olive case, including the importance of early prediction of olive yield, could be incorporated into the case study section (or study area). Second, the review of key factors in crop yield prediction needs to be organized to list variables and their related indices more succinctly. Integrating this information would enhance the informativeness of Table 2.

2.2 In line 178, a typo: ‘stydy’

2.3 Creating a table for three numbers is unnecessary (Table 1)

3. Data and method

3.1 In line 346, ‘in which only those plots of olive trees have been considered’, Does this imply that several cities (municipalities) are selected from all 97 municipalities here? Please provide clarification and mapping details.

3.2 In line 347, it mentioned ‘process the spectral images’, what kind of information has been collected from Google Earth Engine? The polygons in each city or others?

3.3 In line 349, what is simplification process? More details should be provided regarding this geoprocessing (remove line strings in a defined polygon in Figure 2 ?).

3.4 For line 360 to 372, why is the olive growing cycle discussed in the dataset instead of incorporating relevant literature to establish a connection with the study's objective, such as early and late prediction?

3.5 In subsection 3.2.2 Weather information, why is the weather variable, sourced from satellite data, treated separately from the satellite data discussed in section 3.2.3 ? Any specific intention? Furthermore, please explain how the three labels mentioned from line 435 to 443 are derived based on temperature and precipitation. Another concern is the presentation of Temperature and Rainfall data in 3.2.3, which requires clarification (different information or other intentions).

3.6 In section 3.2.3 satellite data, the focus should be more on the indices utilized rather than on not using them in this study. Therefore, consolidating all indices and their respective data sources into one table can enhance clarity in this section.

3.7 In table 6, the MSI is not included in Table 5.

3.8 In line 528, what is ECMWF (have not found any clues through the whole manuscript)?

4. Model

4.1 In Initial data preparation, any results or any presentation for each distribution of the variable (in result section)? Or can a statistical description of all independent variables and dependent variables be provided in a Table?

4.2 In line 567, what kind of algorithm was used in anomaly identification?

4.3 In data transformation, does it refer to the standardization or normalization of the data? If so, what specific technique is being referred to? Please provide the equation for the technique mentioned.

4.4 In line 581, what is the ‘unify all these data into a single source’?

4.5 Does the point 7 mean the feature selection? What is the meaning of ‘we also check different ways of measuring error’? using different indices?

4.6 The 8 points can be aggregated to workflow process subsection by providing more details of model/algorithms used in each step. Section 3.3 can be organized flowing the workflow processing orders.

4.7 From line 609 to 620, please clarify how SVM has been applied to AD. If so, is it employed for classification, and if it is, is it binary or multi-class classification? Additionally, which type of outliers will be eliminated in this context? Alternatively, is this technique referring to unsupervised one-class SVM? Another concern is the intention of SVM used for AD in this dataset? It also indicates that the performance of GLM is worse compared to SVM; to what degree is this disparity?

4.8 What kind of index (why the value of feature above 1 is selected) is used and how the MDL help to select the features? In Line 630, it mentioned model selection which need more specification.

4.9 What is the meaning of ‘the restrictions only depend on the limitations imposed by the hardware used’ in line 659?

4.10 In line 671, what is the intention of categorization? Does this mean that it transitions from a regression task to a classification task?

4.11 From line 680 to 683, it should be more specific while you use the MAE, the n is the sample numbers, so if the prediction for one year is based on 97 cities (the geospatial unit of analysis should also be mentioned in the study area), so the n is 97. If you implement 10 times prediction for 10 years, you will get 10 MAEs here.

4.12 In line 690, what is the value of k?

4.13 The hyperparameter processing SVM is missing, i.e., selection the optimised parameters?

4.14 The olive oil is litres in Figure 3 but Kg in other places in the manuscript.

4.15 A concern is that what is the mathematic representation of VI index for a city-level as an input in the SVM model? A vector or a matrix? Please make more specific for each variable input and the sample numbers (i.e., cities and annual months) in the beginning of the workflow introduction.

5. Results and discussion

5.1 It is preferable to create a plot depicting the points based on predicted values and actual values. This approach facilitates the generation of metrics such as R-square (RMSE and MSE can be considered as well). Furthermore, why MAE is solely showed in the training set and not the testing set as it mentioned trained using data from seven years to predict outcomes for another year in line 697.

Another concern of MAE is the type of value utilized for calculation, whether it's the normalized value (ranging from -1 to 1) or the actual value (potentially 1000,000 Kg of olive crop). Demonstrating the error in actual values is essential for assessing the efficiency and accuracy of the SVM in predicting the olive crop and oil.

5.2 It is confused that line 715 involves predicting the year 2017 using data from 2018 to 2021. Could this be one part of the cross-validation procedure? There is a need for clarification regarding the training (including hyperparameters, cross-validation) and testing processes (alongside which part of data) involved in this case.

5.3 The discussion is brief, particularly regarding the limitations identified in this study, such as issues with the model and data representation.

Reviewer #8: The aim of the study was to predict accurate yield data for olive groves in the province of Jaen in the Andalusia region of southern Spain at the beginning of the growing season using machine learning algorithms and a thorough analysis of the predictor variables.

The outline of the paper is generally appropriate and has some strengths, although some major details need to be revised before acceptance:

1. The introduction (section 1) lacks proper referencing, making it essential to include references in the sentences to support the content effectively.

2. The second part deals well with the subject of theory and also the review of related studies, in this sense this part was well written.

3. The third section of the article elucidates the study area, providing comprehensive details on data collection and corresponding analysis, presented in a well-crafted manner.

4. The section on the results is also acceptable, as it clearly presents the results in two subsections - prediction based on meteorological variables and prediction based on the vegetation index.

5. Nevertheless, it is essential to present the discussion section clearly in order to enable a thorough examination of the results and their comparison with the findings from other studies. In this way, the unique contributions of the authors in the context of this study should be emphasised.

6. In addition, as can be seen in the conclusion section, the strengthening of the content in relation to the research implication as well as its limitations and suggestions for future studies should be clearly presented.

Reviewer #9: The authors have addressed all the questions and comments, and this manuscript is good to publish. The statistical analysis has been performed well according to the data obtained. The English Language and grammar meet the requirements to get published in Plos One. This research work will be helpful for better understanding of the early season crop yield prediction with machine learning as it is demanding in the industry and its a new approach in this field.

Reviewer #10: The study may focus on the machine learning section

In general, the article should be reviewed in accordance with the journal writing rules.

table 6 needs to be reorganized

Results section should be expanded.

Reviewer #11: I appreciate the authors' effort in applying SVM to predict crop yields using temporal data. However, there are several limitations and areas of improvement that I'd like to highlight:

1. A more detailed comparison with traditional time series analysis methods is necessary. It would be beneficial to include a discussion on the specific advantages of SVM in handling the nonlinearities and complexities of the data involved in this study.

2. The results section primarily relies on point estimates for performance metrics. In real-world scenarios, understanding the uncertainty around these estimates is crucial. The authors should consider providing confidence intervals.

3. For any machine learning study, it's crucial to compare the performance against standard baselines.

4. This work doesn't thoroughly justify the choice of the SVM model for this particular task. Are there specific characteristics of the dataset or problem domain that make SVM particularly suited, or was it an arbitrary choice?

5. SVM has several hyperparameters, such as the regularization constant (C), the kernel type (linear, polynomial, radial basis function, etc.), and parameters specific to those kernels (like degree for polynomial or gamma for RBF). These parameters can significantly affect performance. Not discussing them leaves out crucial aspects of the model's behavior. Also, without knowledge of the chosen hyperparameters, there's the concern of potential overfitting. For example, an SVM with a small value of C might fit the training data very closely, possibly capturing noise and leading to poor generalization on unseen data. To strengthen the paper, the authors should have included details on the SVM parameters, the rationale behind their selection, and potentially even results from a hyperparameter tuning process. This would provide more clarity and confidence in the presented results.

Reviewer #12: 1-The abstract must clearly articulate the originality of the investigation.

2-The authors did not specify the method used to determine the sample size.

3-If the authors draw A flowchart to the workflow in the methodology, the part will help link the audiences to how this method advances current knowledge and will help the audiences follow the results quickly.

4-The researchers need to mention the experimental design they used in this manuscript to help the readers follow the results and the conclusions.

Reviewer #13: 1. Based on the detailed examination of the “Materials and Methods” section and subsequent results and discussion, the manuscript demonstrates a methodologically sound approach to predicting olive crop yields using satellite imagery and machine learning techniques, specifically SVM with Linear and Gaussian kernels. The study incorporates rigorous data preprocessing, anomaly detection, and transformation steps to optimize the input for predictive modeling. It also employs regression and analysis and evaluates the influence of various predictors on the target variables (kg of live fruit and olive oil).

The use of k-fold cross-validation for model accuracy assessment and the detailed analysis of mean absolute error (MAE) across different models and predictor sets further illustrate the rigorous statistical and experimental methods applied. The results section substantiates the effectiveness of integrating meteorological variables and vegetation indices into the predictive model, highlighting improvements in prediction accuracy through these methods.

Given these observations the manuscript can be considered partly technically sound as it follows rigorous experimental procedures, employs appropriate controls (e.g., cross-validation), and uses relevant sample sizes for the machine learning model development and validation process. The conclusions drawn from the data presented are supported by the analysis conducted, indicating that the study’s outcomes are rooted in the data obtained and processed through the described methodologies.

However, the assessment of “partly” also acknowledges that while the manuscript demonstrates technical rigor in its approach, the effectiveness and robustness of the predictive models could benefit from further validation across broader datasets or independent validation sets to strengthen the conclusions drawn. Additionally, discussions on the limitations of the study, potential biases in the data or model, and the generalizability of the findings to other regions or crop types would enhance the manuscript’s depth and scientific robustness.

2. Based on the review of the "Materials and Methods" section and related content, including the statistical methods applied for data analysis and model evaluation, the statistical analysis in the manuscript has been performed appropriately and rigorously.

3. Based on the information provided in the “Materials and Methods” and other relevant sections of the manuscript, the authors have indicated that the primary data sources for their study are publicly accessible, including datasets form regional governmental institutions and publicly available web servers for meteorological data. The use of Google Earth Engine for processing satellite images is also mentioned, suggesting the data used in the study are accessible through public platforms or repositories. However, the manuscript does not provide explicit details on accessing these datasets, such as direct links, access instructions, or deposit information in a public repository specific to the study’s datasets.

Given the lack of explicit instructions or links for direct data access or the deposit of processed data in a recognized public repository, and in the absence of a clearly stated Data Availability Statement that meets the PLOS Data policy’s requirements for unrestricted access to all underlying data, the answer would be “No”. The manuscript does not fully comply with the requirements to make all data underlying the findings fully available without restrictions, as per the PLOS Data policy guidelines.

4. Based on the sections of the manuscript reviewed, including the abstract, introduction, materials and methods, results and conclusions, the manuscript is presented in an intelligible fashion and is written in standard English. The test maintains a formal and scientific tone, appropriate for an academic audience, with technical terms and concepts explained in a manner that should be accessible to readers familiar with the field. There are no significant typographical or grammatical errors noted in the sections reviewed, and the manuscript follows a logical structure that facilitates understanding of the study’s objectives, methods, results and implications.

6. PLOS authors have the option to publish the peer review history of their article (what does this mean?). If published, this will include your full peer review and any attached files.

Reviewer #1: No

Reviewer #2: **Yes: **Noe Aguilar Rivera

Reviewer #3: No

Reviewer #4: **Yes: **Kehinde Adewole Adeboye, Ekiti State Polytechnic, Isan Ekiti, Nigeria

Reviewer #5: No

Reviewer #6: **Yes: **Muhammad Ahmad Pasha

Reviewer #7: No

Reviewer #8: **Yes: **Sina Ahmadi Kaliji

Reviewer #9: **Yes: **Muhammad Afaq Ahmed

Reviewer #10: No

Reviewer #11: No

Reviewer #12: No

Reviewer #13: No

---

## [Author Response · Author response to Decision Letter 0]

27 Aug 2024

We have attached a file with the response to each reviewer

---

## [Decision Letter · Decision Letter 1]

20 Sep 2024

Improving early prediction of crop yield in Spanish olive groves using satellite imagery and machine learning

PONE-D-24-04068R1

Dear Dr. Isabel Ramos

We’re pleased to inform you that your manuscript has been judged scientifically suitable for publication and will be formally accepted for publication once it meets all outstanding technical requirements.

Kind regards,

Salim Heddam

Academic Editor

PLOS ONE

Reviewers' comments:

Reviewer's Responses to Questions

**Comments to the Author**

1. If the authors have adequately addressed your comments raised in a previous round of review and you feel that this manuscript is now acceptable for publication, you may indicate that here to bypass the “Comments to the Author” section, enter your conflict of interest statement in the “Confidential to Editor” section, and submit your "Accept" recommendation.

Reviewer #2: All comments have been addressed

Reviewer #5: All comments have been addressed

Reviewer #8: (No Response)

Reviewer #12: (No Response)

2. Is the manuscript technically sound, and do the data support the conclusions?

Reviewer #2: Yes

Reviewer #5: Yes

Reviewer #8: (No Response)

Reviewer #12: (No Response)

3. Has the statistical analysis been performed appropriately and rigorously? 

Reviewer #2: Yes

Reviewer #5: Yes

Reviewer #8: (No Response)

Reviewer #12: (No Response)

4. Have the authors made all data underlying the findings in their manuscript fully available?

Reviewer #2: Yes

Reviewer #5: Yes

Reviewer #8: (No Response)

Reviewer #12: (No Response)

5. Is the manuscript presented in an intelligible fashion and written in standard English?

Reviewer #2: Yes

Reviewer #5: Yes

Reviewer #8: (No Response)

Reviewer #12: (No Response)

6. Review Comments to the Author

Reviewer #2: (No Response)

Reviewer #5: All my comments have been considered. Authors have exerted great effort to enhance the paper quality, I

recomend to accept the paper for publishing.

Reviewer #8: The authors effectively incorporated my comments and successfully addressed the suggested points in their revised article.

Reviewer #12: (No Response)

7. PLOS authors have the option to publish the peer review history of their article (what does this mean?). If published, this will include your full peer review and any attached files.

Reviewer #2: No

Reviewer #5: No

Reviewer #8: **Yes: **Sina Ahmadi Kaliji

Reviewer #12: No

---

## [Editor Report · Acceptance letter]

31 Oct 2024

PONE-D-24-04068R1 

PLOS ONE

Dear Dr. Ramos, 

I'm pleased to inform you that your manuscript has been deemed suitable for publication in PLOS ONE. Congratulations! Your manuscript is now being handed over to our production team.

Kind regards, 

on behalf of

Dr. Salim Heddam 

Academic Editor

PLOS ONE